# Individual Welfare Guarantees in the Autobidding World with Machine-learned Advice

## ABSTRACT

Online advertising channels commonly focus on maximizing total advertiser welfare to enhance channel health, and previous literature has studied augmenting ad auctions with machine learning predictions on advertiser values (also known as *machine-learned advice*) to improve total welfare. Yet, such improvements could come at the cost of individual bidders' welfare and do not shed light on how particular advertiser bidding strategies impact welfare. Motivated by this, we present an analysis on an individual bidder's welfare loss in the autobidding world for auctions with and without machine-learned advice, and also uncover how advertiser strategies relate to such losses. In particular, we demonstrate how ad platforms can utilize ML advice to improve welfare guarantee on the aggregate and individual bidder level by setting ML advice as personalized reserve prices when the platform consists of *autobidders* who maximize value while respecting a return on ad spend (ROAS) constraint. Under parallel VCG auctions with such ML advice-based reserves, we present a worst-case welfare lower-bound guarantee for an individual autobidder, and show that the lower-bound guarantee is positively correlated with ML advice quality as well as the scale of bids induced by the autobidder's bidding strategies. Further, we show that no truthful, and possibly randomized mechanism with anonymous allocations can achieve universally better individual welfare guarantees than VCG, in the presence of personalized reserves based on ML-advice of equal quality. Moreover, we extend our individual welfare guarantee results to generalized first price (GFP) and generalized second price (GSP) auctions. Finally, we present numerical studies using semi-synthetic data derived from ad auction logs of a search ad platform to showcase improvements in individual welfare when setting personalized reserve prices with ML-advice.

**ACM Reference Format:**
Anonymous Author(s). 2024. Individual Welfare Guarantees in the Autobidding World with Machine-learned Advice. In *Proceedings of ACM Conference (Conference'17)*. ACM, New York, NY, USA, 21 pages. https://doi.org/XXXXXXX.XXXXXXX

## 1 INTRODUCTION

Online advertisers have access to a vast array of digital advertising channels, such as social media, web display, and keyword search, from which they can procure ad impressions and drive user traffic.

One possible way for these channels to improve overall attractiveness and retention is to design appropriate ad auction mechanisms that enhance advertisers' total welfare, which reflects the aggregate advertiser-perceived value of procured ad impressions on the channel. For instance, consider advertisers whose ad campaign objective is to maximize ad clicks that direct users to landing pages of their services or products, as described in [23]. These advertisers' perceived value of procured ad impressions is their click conversion rate, and thereby ad channels' welfare maximization goal translates into improving the aggregate realized click conversion among all participating advertisers.

Academic literature has developed various approaches to improve total welfare, one of which involves using machine learning tools to predict advertiser values based on user interactions with ads. In the instance where welfare corresponds to click conversion, channels use ML algorithms to produce predictions (i.e., ML advice) on click conversion rates for impressions. See [36, 42, 44] or [48] for a comprehensive survey on click predictions. Having obtained ML advice on advertiser values, recent works such as [3, 15, 16] motivate the approach to augment existing ad auctions by directly setting personalized reserve prices for advertisers using such ML advice, and show theoretical guarantees on total welfare improvement.

Nevertheless, these results present two important issues. First, improving total welfare doesn't necessarily ensure that all individual advertisers benefit equally; some may even experience a detriment to their welfare. For instance, larger advertisers acquiring more impressions while smaller advertisers receive fewer could potentially harm the businesses of the latter and compromise the overall health of the channel in the long term. Second, welfare improvement guarantees are presented in a price of anarchy (POA) fashion, which measures the worst-case total welfare outcome compared to the maximum achievable (or efficient) welfare. However, these POA bounds are independent of advertiser bidding strategies and thus don't illuminate how specific advertiser bidding strategies in ad auctions affect individual or total welfare. Given these shortcomings in existing results, in this work, we address the following questions:

*Given an advertiser's bidding strategy to procure impressions in ad auctions, how can platforms characterize the potential welfare loss for this individual advertiser? How should ad channels utilize machine-learned advice that predicts advertiser values to improve individual welfare?*

We study a prototypical *autobidding setting* where advertisers compete simultaneously in numerous multi-slot position auctions that are run in parallel, and aim to maximize total advertiser value under *return-on-ad-spend (ROAS)* constraints that restrict total spend of a bidder to be less than her total acquired value across all auctions in an average sense; see similar setups in [1, 3, 16, 37]. On the other hand, ad platforms possess ML advice that predicts

advertisers' real values with a certain degree of accuracy/quality. Under this setup, our main contributions are described as followed:

**Strategy-dependent individual welfare guarantee metric for individual advertisers.** In Section 2, we present a novel individual welfare metric that measures the difference between two specific welfare outcomes for an individual advertiser: (1) given a fixed bidding strategy, the worst-case welfare across all auction outcomes where all bidders' ROAS constraints are satisfied; and (2) the welfare that this individual bidder would have obtained in the global welfare-maximizing outcome. Our metric achieves two key goals: (1) it characterizes individual welfare loss, and (2) it allows platforms to uncover the relationship between advertiser strategies and individual welfare guarantees.

**Individual welfare guarantees in VCG auctions with ML-advice-based personalized reserves.** In Section 3, we illustrate through examples that setting ML advice as personalized reserves, as presented in [3, 15, 16], improves individual welfare guarantees under our individual welfare metric. In Section 4, we demonstrate that augmenting VCG auctions with ML-advice-based reserves enables us to present an individual welfare lower bound guarantee that increases with the advertiser's bid scale, quality of ML advice, and the relative market share of this advertiser compared to competitors (Theorem 4.1). Together with the results in [16], we conclude that incorporating ML advice as personalized reserves achieves a "best of both worlds" outcome by simultaneously benefiting total and individual welfare.

**VCG has the best individual welfare guarantees among a broad class of auctions.** In Section 5, we demonstrate that no allocation-anonymous, truthful, and possibly randomized auction format with ML advice of a given quality can surpass the individual welfare guarantee provided by the VCG auction combined with ML advice of the same quality; see Theorem 5.1. Specifically, for any allocation-anonymous, truthful, and potentially randomized auction, we construct a problem instance with personalized reserves based on ML advice of the specified quality and show that there must be at least one bidder whose welfare does not exceed the welfare lower bound guarantee under VCG (refer to Theorem 4.1).

**Extending individual welfare guarantees to GSP and GFP.** We extend the individual welfare guarantee results to GSP and GFP auctions and demonstrate that a similar individual welfare lower bound guarantee for VCG continues to hold (see Theorem 6.2). We compare these lower bound guarantees in GSP and GFP with those of VCG and identify conditions under which VCG either outperforms or underperforms GSP/GFP in terms of our individual welfare metric, given the same ML advice quality.

**Numerical results.** We present numerical studies using semi-synthetic data derived from the auction logs of a search ad platform to showcase individual welfare improvement by setting ML-advice-based personalized reserves. We demonstrate that as the quality of ML advice improves, the welfare of more advertisers approaches what they would have obtained in the efficient outcome.

## 1.1 Related works

**Autobidding and total welfare maximization.** The works most relevant to this paper are [3, 16, 37], where they consider the same autobidding setting (i.e., value-maximizers with ROAS constraints)

as ours. [3, 15, 16, 37] all present techniques to improve the price-of-anarchy bounds for the total welfare of any feasible outcome in which all bidders' ROAS constraints are satisfied. [16] relies on additive boosts to bid values, [3, 15] utilizes approximate reserve prices derived from ML advice, and [37] develops randomized allocation and payment rules. Our work distinguishes itself from these works as we focus on welfare guarantees at the individual bidder level, and also shed light on how autobidders' uniform bidding strategies influence individual welfare loss. We note that our proof techniques also differ from those in [3, 15, 16, 37] as our individual welfare guarantees require novel analyses on the value-expenditure trade-offs that individual bidders would face when tempted to outbid others to acquire more value. See the discussion in Section 4 for more details.

**Exploiting ML advice.** ML advice has been utilized in various applications to enhance decision-making. For example, [47] exploits ML advice to develop algorithms for the multi-shop ski-rental problem, [35] adopts ML advice for the caching problem, [30] studies online page migration using ML advice, and [27] studies online resource allocation with convex ML advice. However, even though numerous works in online advertising have examined predictive models for advertiser values, click-through rates, etc. (see, e.g., [34, 42, 44]), the literature on applying such predictions (or more generally, ML advice) to mechanism design problems remains limited. Also, refer to [11, 26] for works that exploit sample information (unstructured ML advice) in online decision-making. One pertinent study in this realm is [38], which devises a theoretical framework to optimize reserve prices in a posted price mechanism by leveraging prediction inputs on bid values. Unlike this research, we do not focus on optimizing reserves but advocate for the straightforward approach of setting reserves using ML advice to enhance individual advertiser welfare. Furthermore, we emphasize that our work contributes to the field by harnessing ML advice to design mechanisms that bolster welfare guarantees for individual bidders.

We refer readers to Appendix A for a further literature review.

## 2 PRELIMINARIES

**Auction Model with Advertisers and Position Auctions.** Consider $N$ bidders (i.e., advertisers) participating in $M$ parallel position auctions $(\mathcal{A}_j)_{j \in [M]}$, where each auction $\mathcal{A}_j$ is instantiated by a user keyword search query. An auction $\mathcal{A}_j$ offers $L_j \geq 1$ ad slots to bidders. These slots are ordered by visual prominence, or equivalently, the likelihood of the user viewing the slot on the webpage. This likelihood is represented by click-through-rates (CTR): $1 \geq \mu_j(1) \geq \mu_j(2) \geq \ldots \geq \mu_j(L_j) \geq 0$. Here, $\mu_j(\ell)$ denotes the likelihood of the user in auction $j$ viewing slot $\ell \in [L_j]$ (for an introduction to position auctions, see, e.g., [19, 33, 46]). A bidder $i \in [N]$ has a private value-per-click, denoted by $v_{i,j} > 0$, for auction $\mathcal{A}_j$. This value represents her perceived value conditioned on the user viewing her ad. Consequently, if she wins slot $\ell \in [L]$, her accrued welfare or value is $\mu_j(\ell) \cdot v_{i,j}$.

## 2.1 Preliminaries for a single position auction

In this subsection, we discuss a single position auction, and thus temporarily remove subscripts in auction indices $j$. A (possibly randomized) position auction $\mathcal{A}$ with $L \geq 1$ slots is characterized by a tuple $(\mathcal{X}, \mathcal{P}, \boldsymbol{\mu})$, where $\mathcal{X}$ is an allocation rule, $\mathcal{P}$ is a payment

rule, and CTRs $\boldsymbol{\mu} = (\mu(\ell))_{\ell \in [L]} \in [0, 1]^L$ that satisfies $1 \geq \mu(1) \geq \mu(2) > \ldots \geq \mu(L) \geq 0$. Let $N$ bidders with private value per-clicks $\boldsymbol{v} = (v_i)_{i \in [N]}$ participate in auction $\mathcal{A}$ by submitting a bid profile $\boldsymbol{b} = (b_i)_{i \in [N]} \in \mathbb{R}_+^N$, and we describe the payment and allocation rules as follows.

The allocation rule $\mathcal{X} : \mathbb{R}_+^N \to \{0, 1\}^{N \times L}$ maps bid profile $\boldsymbol{b} \in \mathbb{R}_+^N$ to an outcome $\boldsymbol{x} = \mathcal{X}(\boldsymbol{b}) \in \{0, 1\}^{N \times L}$ which may possibly be random. The entry $x_{i,\ell} = 1$ if bidder $i$ is allocated slot $\ell \in [L]$, and 0 otherwise. Here, each slot $\ell$ is at most allocated to one bidder so $\sum_{i \in [N]} x_{i,\ell} \leq 1$ for any $\ell$. Further, under outcome $\boldsymbol{x} \in \{0, 1\}^{N \times L}$, bidder $i$ who has value $v_i$ attains a total welfare of $W_i(\boldsymbol{x}) = v_i \sum_{\ell \in [L]} \mu(\ell) x_{i,\ell}$. That is, if bidder $i$ is allocated slot $\ell \in [L]$ (i.e., $x_{i,\ell} = 1$), her welfare is $W_i(\boldsymbol{x}) = \mu(\ell) v_i$. The payment rule $\mathcal{P} : \mathbb{R}_+^N \to \mathbb{R}_+^N$ maps bids $\boldsymbol{b}$ to payments $\mathcal{P}(\boldsymbol{b}) \in \mathbb{R}_+^N$ where $\mathcal{P}_i(\boldsymbol{b})$ is the payment of bidder $i$. In this work, we focus on the class of auctions that are *ex-post individual rational* (IR), i.e. the payment for any bidder is less than her submitted bid, or mathematically $\mathcal{P}_i(b_i, \boldsymbol{b}_{-i}) \leq b_i$ for any $\boldsymbol{b}_{-i} \in \mathbb{R}_+^{N-1}$.[1] We note that the classic VCG, GSP and GFP auctions are ex-post IR.

In the following we define truthful, allocation anonymous auctions and personalized reserve augmented auctions.

**Definition 2.1** (Truthful auction). *Consider the position auction $\mathcal{A} = (\mathcal{X}, \mathcal{P}, \boldsymbol{\mu})$ where, recall that $\mathcal{X}$ and $\mathcal{P}$ are possibly random allocation and payment rules, respectively, and $\boldsymbol{\mu} \in [0, 1]^L$ represents CTRs. We say the auction is truthful if, for any bidder $i \in [N]$, all of their values $v_i$ are such that $v_i \in \arg\max_{b \geq 0} \mathbb{E}[W_i(\mathcal{X}(b, \boldsymbol{b}_{-i})) - \mathcal{P}_i(b, \boldsymbol{b}_{-i})]$ for any competing bid profile $\boldsymbol{b}_{-i}$. Here, the expectation is taken with respect to the possible randomness in $(\mathcal{X}, \mathcal{P})$, and recall that the welfare $W_i(\boldsymbol{x}) = v_i \sum_{\ell \in [L]} \mu(\ell) x_{i,\ell}$ with $\boldsymbol{x} = \mathcal{X}(b, \boldsymbol{b}_{-i})$.*

Note that the well-known VCG auction is truthful. In truthful auctions, it is a weakly dominant strategy for a bidder to bid her true value when her objective is to maximize quasi-linear utility. In the next Subsection 2.2, we study bidders whose objectives are not necessarily quasi-linear, meaning that truthful bidding is no longer weakly optimal in truthful auctions.

We next define allocation-anonymous auctions. In such auctions, if two bidders swap their bids, the probability of each bidder winning any slot will also be swapped. In other words, the outcome of the position auction depends solely on the solicited bid values and is independent of the bidders' identities. The classic VCG, GSP, and GFP are all allocation-anonymous. We refer readers to Example B.1 in the appendix for an illustrative example of allocation-anonymity in GSP.

**Definition 2.2** (Allocation anonymous auctions). *Given a fixed position auction $\mathcal{A} = (\mathcal{X}, \mathcal{P}, \boldsymbol{\mu})$, consider any permutation $\sigma : [N] \to [N]$ of $\{1, \ldots, N\}$ and the permuted bid profile $\boldsymbol{b}' = (b_{\sigma(i)})_{i \in [N]}$. Let $\boldsymbol{x} = \mathcal{X}(\boldsymbol{b})$ and $\boldsymbol{x}' = \mathcal{X}(\boldsymbol{b}')$ represent the (possibly random) outcomes under $\boldsymbol{b}$ and $\boldsymbol{b}'$, respectively. Then, $\mathcal{X}$ is termed allocation-anonymous if, for any bidder $i \in [N]$ and slot $j \in [L]$, we have $\mathbb{P}(x_{\sigma(i),j} = 1 | \boldsymbol{b}) = \mathbb{P}(x'_{i,j} = 1 | \boldsymbol{b}')$.*

Finally, we describe augmenting allocation anonymous auctions with personalized reserves.

**Definition 2.3** (Personalized-reserve augmented allocation anonymous auctions). *Fix position auction $\mathcal{A} = (\mathcal{X}, \mathcal{P}, \boldsymbol{\mu})$, and some vector of personalized reserve prices $\boldsymbol{r} \in \mathbb{R}_+^N$ where $r_i$ is the reserve price for bidder $i \in [N]$. Then, the augmented auction is $\mathcal{A}' = (\mathcal{X}', \mathcal{P}', \boldsymbol{\mu})$ where allocation $\mathcal{X}'$ and payment $\mathcal{P}'$ are characterized via the following procedure for any bid profile $\boldsymbol{b} \in \mathbb{R}_+^N$:*

- *$\mathcal{X}'$: Define bid profile $\boldsymbol{b}' = (b_i \cdot \mathrm{I}\{b_i \geq r_i\})_{i \in [N]}$. Then $\mathcal{X}'(\boldsymbol{b}) = \mathcal{X}(\boldsymbol{b}')$.[2]*
- *$\mathcal{P}'$: If $i \in [N]$ is not allocated a slot under outcome $\mathcal{X}'(\boldsymbol{b})$, $\mathcal{P}'_i(\boldsymbol{b}) = 0$. Otherwise, let $\ell_i \in [L]$ be the slot allocated to bidder $i$ under $\mathcal{X}'(\boldsymbol{b})$. Then, $\mathcal{P}'_i(\boldsymbol{b}) = \max\{\mathcal{P}_i(\boldsymbol{b}'), \mu(\ell_i) \cdot r_i\}$.*

Recall that a bid of 0 always results in neither allocation nor payment. Therefore, $\mathcal{X}'$ can effectively be viewed as excluding bidders who don't meet their reserves and then implementing the allocation rule $\mathcal{X}$ for the remaining bidders. In subsequent sections, we'll demonstrate that the personalized reserve prices pertinent to this work, which are based on ML-advice, ensure all bidders clear their reserves, implying that no bidders will be excluded from ranking. For an illustration of how anonymous VCG, GSP, and GFP auctions are augmented with personalized reserves, we refer readers to Example B.2 in the appendix.

## 2.2 Autobidders' objectives and uniform bidding strategies

Recall the setup with $N$ bidders participating in $M$ parallel position auctions $(\mathcal{A}_j)_{j \in [M]}$, where $\boldsymbol{v}_j \in \mathbb{R}_+^N$ are the bidders' values in auction $\mathcal{A}_j = (\mathcal{X}_j, \mathcal{P}_j, \boldsymbol{\mu}_j)$ (refer to definitions in Subsection 2.1).

*Notations:* The bids submitted by bidder $i$ are denoted by $\boldsymbol{b}_i \in \mathbb{R}_+^M$. The bid profile submitted to $\mathcal{A}_j$ is represented by $\boldsymbol{b}_j^\top \in \mathbb{R}_+^N$. The outcome of $\boldsymbol{b}_j^\top$ in $\mathcal{A}_j$ is given by $\mathcal{X}_j(\boldsymbol{b}_j^\top) \in \{0, 1\}^{N \times L_j}$. The payment vector in $\mathcal{A}_j$ is $\mathcal{P}_j(\boldsymbol{b}_j^\top) \in \mathbb{R}_+^N$. Whether bidder $i$ wins slot $\ell$ in $\mathcal{A}_j$ is indicated by $\mathcal{X}_{i,\ell,j}(\boldsymbol{b}_j^\top) \in \{0, 1\}$. The payment of bidder $i$ in $\mathcal{A}_j$ is $\mathcal{P}_{i,j}(\boldsymbol{b}_j^\top) \in \mathbb{R}_+$. Lastly, $\mathcal{X}(\boldsymbol{b})$ is defined as $(\mathcal{X}_j(\boldsymbol{b}_j^\top))_{j \in [M]}$, representing the collection of all auction outcomes.

**Bidder welfare and ROAS constraints.** The welfare of bidder $i$ in auction $\mathcal{A}_j$ is denoted by $W_{i,j}(\mathcal{X}_j(\boldsymbol{b}_j^\top))$. Her total welfare across all auctions is given by $W_i(\mathcal{X}(\boldsymbol{b}))$. These are formally defined as

$$W_i(\mathcal{X}(\boldsymbol{b})) := \sum_{j \in [M]} W_{i,j}(\mathcal{X}_j(\boldsymbol{b}_j^\top))$$

$$\text{and} \quad W_{i,j}(\mathcal{X}_j(\boldsymbol{b}_j^\top)) = \sum_{\ell=1}^{L_j} \mu_j(\ell) \cdot v_{i,j} \cdot \mathcal{X}_{i,\ell,j}(\boldsymbol{b}_j^\top). \quad (1)$$

Each bidder is subject to a *return-on-ad-spend (ROAS)* constraint. This mandates that her total expenditure across all auctions be

---

[1]ex-post IR does not require payment being at most value. In other words, the payment in a single auction can exceed value, and when this occurs for an advertiser, the advertiser would need to cover such a loss by winning other auctions with value larger than payment.

[2]This allocation is known as an *eager implementation* of personalized reserve prices, where any high-ranked slots are always allocated before a lower-rank slot gets allocated. There also exists a *lazy implementation*, where we first rank all bids, and then allocate slots to each bidder following this ranking if the bidder clears his reserve. Note that the lazy implementation may leave "holes" in allocation, e.g., the first and third slots are allocated while leaving the second slot un-allocated. It will become clear later that all results in this work hold for both the eager and lazy implementation of personalized reserve prices.

less than her total acquired value.[3] Mathematically, for a fixed competing bid profile $\boldsymbol{b}_{-i} \in \mathbb{R}_+^{(N-1) \times M}$, the ROAS constraint for bidder $i$ is

$$\mathbb{E}\left[W_i(\mathcal{X}(\boldsymbol{b}_i, \boldsymbol{b}_{-i}))\right] \geq \mathbb{E}\left[\mathcal{P}_i(\boldsymbol{b}_i, \boldsymbol{b}_{-i})\right]$$

$$\text{where } \mathcal{P}_i(\boldsymbol{b}_i, \boldsymbol{b}_{-i}) := \sum_{j \in [M]} \mathcal{P}_{i,j}(\boldsymbol{b}_j^\top). \qquad (2)$$

The expectation here is with respect to possible randomness in the allocation and payment rules of auctions $(\mathcal{A}_j)_{j \in [M]}$. When allocation and payment for $(\mathcal{A}_j)_{j \in [M]}$ are deterministic (like in VCG, GSP, and GFP), the expectation is omitted for clarity.

**Definition 2.4** (Feasible bid profiles). *For parallel auctions $(\mathcal{A}_j)_{j \in [M]}$, a bid profile $\boldsymbol{b} \in \mathbb{R}_+^{N \times M}$ is feasible if Eq. (2) is satisfied for all bidders. All feasible bid profiles are denoted by $\mathcal{F}$. Given a bidder $i$ and her bids $\boldsymbol{b}_i \in \mathbb{R}_+^M$, $\mathcal{F}_{-i}(\boldsymbol{b}_i) = \{\boldsymbol{b}_{-i} \in \mathbb{R}_+^{(N-1) \times M} : (\boldsymbol{b}_i, \boldsymbol{b}_{-i}) \in \mathcal{F}\}$.*

In simpler terms, $\mathcal{F}_{-i}(\boldsymbol{b}_i)$ captures all competing bid profiles for bidder $i$ that ensure all bidders' ROAS constraints are met.

**Autobidder and uniform bidding.** A bidder is termed an *autobidder* if she aims to maximize the welfare $\mathbb{E}\left[W_i(\mathcal{X}(\boldsymbol{b}_i, \boldsymbol{b}_{-i}))\right]$ subject to the ROAS constraint in Eq. (2). Autobidding represents advertisers' behavior of maximizing conversions while adhering to constraints on spend. For any autobidder $i$, Proposition B.1 in the appendix demonstrates that the optimal bidding strategy in truthful auctions, given any competing bid profile, is *uniform bidding*, where the submitted bids are $\boldsymbol{b}_i = \alpha_i \boldsymbol{v}_i$ with some *uniform bid multiplier* $\alpha_i \geq 1$.

## 2.3 Efficient auction outcomes and individual welfare guarantees

Let $\ell_{i,j}^*$ be the ranking of bidder $i$ in auction $\mathcal{A}_j$, based on decreasing order of true values $\boldsymbol{v}_j \in \mathbb{R}_+^N$. We define the outcome $\boldsymbol{x}^* = (x_j^*)_{j \in [M]}$ with $x_{i,\ell,j}^* = \mathbb{I}\{\ell = \ell_{i,j}^*\}$ as the *efficient outcome*. This outcome yields the largest total welfare because the allocation of slots in each auction follows the ranking of bidders' true values. In analogy with our welfare definition in Eq. (1), we have

$$\text{OPT}_{i,j} = \mu_j(\ell_{i,j}^*) \cdot v_{i,j}, \qquad \text{OPT}_i = \sum_{j \in [M]} \text{OPT}_{i,j}, \qquad (3)$$

and $\text{OPT} = \sum_{i \in [N]} \text{OPT}_i$. Here, $\text{OPT}_{i,j}$, $\text{OPT}_i$, and $\text{OPT}$ represent the welfare of bidder $i$ in auction $j$, the total welfare contribution of bidder $i$, and the overall total welfare under the efficient outcome, respectively. We adopt the convention that $\mu_j(\ell) = 0$ for all $\ell > L_j$. It's important to recall that the welfare here refers to the bidder's total value, not the surplus (the difference between value and payment). This is because autobidders are primarily interested in the total value they can obtain.

Due to the presence of ROAS constraints, autobidders may adopt arbitrary strategies to optimize personal welfare. This can cause real auction outcomes to deviate from the efficient outcome, resulting in significant welfare losses for some bidders compared to the welfare they would have attained under the efficient one. Conversely, some other bidders may be significantly better off. It is thus important for auction platforms to: (1) provide welfare guarantees on the

individual level; and (2) understand how individual welfare relates to advertiser strategies. In the following Definition 2.5, we present an individual welfare metric that achieves these two goals.

**Definition 2.5** (Individual welfare metric). *Fix a bidder $i \in [N]$ and her bids $\boldsymbol{b}_i \in \mathbb{R}_+^M$. Then the individual welfare metric for $i$ is given by $\min_{\boldsymbol{b}_{-i} \in \mathcal{F}_{-i}(\boldsymbol{b}_i)} \frac{\mathbb{E}[W_i(\mathcal{X}(\boldsymbol{b}))]}{\text{OPT}_i}$, where $\mathcal{F}_{-i}(\cdot)$ is as defined in Definition 2.4. The total welfare $W_i$ under outcome $\mathcal{X}(\boldsymbol{b})$ is as defined in Eq. (1), and the expectation is taken with respect to possible randomness in the auctions.*

In words, our individual welfare metric provides a quantitative answer to the following question: fixing a bid profile $\boldsymbol{b}_i$ for bidder $i$, among all outcomes induced by competing bid profiles $\boldsymbol{b}_{-i} \in \mathcal{F}_{-i}(\boldsymbol{b}_i)$ that ensure every bidder's ROAS constraint is satisfied (see Definition 2.4), what proportion of the welfare under the efficient outcome can be retained under the worst case outcome?

# 3 ML ADVICE FOR BIDDER VALUES AS PERSONALIZED RESERVE PRICES

In this section, we aim to present lower bound guarantees for the individual welfare metric as per Definition 2.5 to ensure each individual autobidder is well off. In particular, we motivate the approach to set personalized reserve prices based on ML advice that takes the form of *lower-confidence bound* predictions on true advertiser values. Note that with modern machine learning (ML) models, it is common for ad platforms to utilize available historical data (e.g. bid logs, keyword characteristics, user profiles, etc.) to produce predictions on autobidders' values; see e.g. [42, 44].

## 3.1 Motivating example

Consider 2 bidders competing in two (single-slot) second-price auctions (i.e. $L_1 = L_2 = 1$) with CTRs $\mu_1(1) = \mu_2(1) = 1$. Suppose that both bidders are autobidders who adopt uniform bidding strategies (see Section 2.2). In particular, suppose that bidder 1 sets her bid multiplier to be $\alpha_1 = 1$. Then, when her competitor, bidder 2, sets a multiplier $\alpha_2 > 2$, bidder 2 will win both auctions and acquire a total value/welfare of $v_{2,1} + v_{2,2} = \frac{3}{2}v$ while submitting a payment of $\alpha_1(v_{1,1} + v_{1,2}) = v$. In this case, bidder 2 satisfies her ROAS constraint, leaving bidder 1 with no value. We also highlight that this bid multiplier profile constitutes an equilibrium because bidder 1 cannot deviate and raise her bid multiplier to outbid bidder 2 for auction 1. With $\alpha_2 > 2$, bidder 1 would violate her ROAS constraint if she bids more than $\alpha_2 v_{2,1} > v$.

Now, suppose that for each value $v_{i,j}$ where $i, j \in$ [2], the platform possesses a ML-based lower-confidence

| | Auction 1 | Auction 2 |
|---|---|---|
| bidder 1 | $v_{1,1} = v$ | $v_{1,2} = 0$ |
| bidder 2 | $v_{2,1} = \frac{v}{2}$ | $v_{2,2} = v$ |

bound, $(\underline{v}_{i,j})_{i,j \in [N]}$, such that for some $\beta > \frac{1}{2}$, $\beta v_{i,j} \leq \underline{v}_{i,j} < v_{i,j}$ for all non-zero $v_{i,j}$, and sets the personalized reserve price $r_{i,j} = \underline{v}_{i,j}$. If bidder 2 attempts to win both auctions by setting $\alpha_2 > 2$, her payment will be at least $\max\{\beta v_{2,1}, \alpha_1 v_{1,1}\} + \max\{\beta v_{2,2}, \alpha_1 v_{1,2}\} = v + \beta v > \frac{3}{2}v$, violating her ROAS constraint. Therefore, by setting personalized reserves with ML advice, bidder 2 is prohibited from outbidding bidder 1 in auction 1, safeguarding bidder 1's welfare.

**Key takeaway from Example 3.1.** From the example, it's evident that without reserve prices, bidder 2 takes advantage of a

---

[3]Our results also apply to more general *return-on-investment (ROI)* constraints. Here, each bidder $i$ has a target ROI $T_i$, and the constraint in Eq. (2) becomes $W_i(\mathcal{X}(\boldsymbol{b}_i, \boldsymbol{b}_{-i})) \geq T_i \cdot \sum_{j \in [M]} p_{i,j}$; see e.g. [13, 25, 29].

significant margin in her ROAS constraint by winning auction 2, where the payment to secure a win is minimal. This allows her to increase her bid and outbid bidder 1 in auction 1 without violating her overall ROAS constraint. Essentially, she offsets the additional cost in auction 1 using the value advantage gained from auction 2.

By setting personalized reserve prices, the platform can raise the payment bidder 2 needs to make in auction 2, thereby reducing her manipulative influence. More specifically, in the absence of reserve prices, bidders who have high aggregate values across auctions might excessively overbid. This allows them to manipulate outcomes in specific auctions by balancing out their costs with the values secured from other auctions. Introducing personalized reserve prices makes this kind of overbidding more costly, and as a result, decreases bidders' overall manipulative power.

## 3.2 Personalized reserve prices using ML advice

Here, we focus on the following notion of *approximate reserve prices* with which we can reduce bidders' manipulative power, as exemplified in Example 3.1, and thereby improve individual welfare.

**Definition 3.1** ($\beta$-accurate ML advice and approximate reserve prices). *Suppose there exists ML advice $(\underline{v}_{i,j})_{i,j \in [N]}$ in the form of a lower-confidence bound. If $\underline{v}_{i,j} \in [\beta v_{i,j}, v_{i,j})$ for any bidder $i \in [N]$ and auction $j \in [M]$ with some $\beta \in (0, 1)$, we say the ML advice is $\beta$-accurate. Further, if the platform sets $r_{i,j} = \underline{v}_{i,j}$, we say reserve prices $r$ are $\beta$-approximate.*

The gap between the lower bound $\beta v_{i,j}$ and the true value $v_{i,j}$ in Definition (3.1) represents the inaccuracies of the platform's ML advice. In other words, $\beta$ can be perceived as a quality measure of the platform's ML advice for advertiser value, such that a larger $\beta$ represents better advice quality.

Furthermore, ML advice in online advertising settings generally concerns predicting advertiser values with historical conversion data and produces confidence intervals of advertiser values (see e.g. [10, 12, 31, 43]). We remark that these confidence intervals can be viewed as a special case of the lower-confidence type of ML advice in Definition (3.1): suppose the platform utilizes some ML model to predict the true value $v_{i,j}$ of bidder $i$ in auction $j$, and produces a confidence interval $(\underline{v}_{i,j}, \bar{v}_{i,j}) \ni v_{i,j}$ with $\underline{v}_{i,j}, \bar{v}_{i,j} > 0$. The platform can then choose a personalized reserve $r_{i,j} = \underline{v}_{i,j}$, which is $\beta$-approximate for $\beta = \underline{v}_{i,j}/\bar{v}_{i,j} \in (0, 1)$ because $\beta v_{i,j} < \beta \bar{v}_{i,j} = \underline{v}_{i,j} = r_{i,j} < v_{i,j}$.

Furthermore, in Definition 3.1, it is assumed that the ML advice $\underline{v}_{i,j}$ is a true lower bound on bidder $i$'s value in auction $j$. This assumption can be relaxed to high probability statements: suppose we possess some prediction $\hat{v}_{i,j}$ for $v_{i,j}$ that satisfies $|\hat{v}_{i,j} - v_{i,j}| < \eta$ with high probability (w.h.p.) for some known $\eta$. Then, the confidence interval $(\hat{v}_{i,j} - \eta, \hat{v}_{i,j} + \eta)$ contains $v_{i,j}$ w.h.p. The platform can then set a personalized reserve $r_{i,j} = \hat{v}_{i,j} - \eta$. With such personalized reserve prices derived from probabilistic ML advice, all results in this paper remain valid w.h.p.

We note that the ML advice accuracy parameter $\beta$ can be considered a lower bound on advertiser value approximations. All our results remain valid if each advertiser's approximation factor is no less than $\beta$. If each bidder $i$ has an approximation factor $\beta_i$, then $\beta$

can represent the minimum among all $\beta_i$. We conclude this section with the following remark.

**Remark 3.1.** *Implementing $\beta$-approximate personalized reserve prices in an allocation-anonymous auction does not impact anonymity. This is because $\beta < 1$ and thus all bidders clear their reserves.*

## 4 INDIVIDUAL WELFARE GUARANTEES FOR VCG WITH ML ADVICE

In the motivating Example 3.1, we observe that ML advice and corresponding $\beta$-approximate reserves allow the parallel auctions to safeguard welfare for individual bidders by increasing payments and consequently limit the manipulative behavior of bidders who face significantly less competition in certain auctions. In this section, through the following Theorem 4.1, we formalize this intuition for the classic VCG auction and present a quantitative measure for the relationship between overall individual welfare and ML advice when incorporated in the form of approximate reserves.

Theorem 4.1 (Lower bound for VCGs with approximate reserves). *Let $(\mathcal{A}_j)_{j \in [M]}$ be VCG auctions, and personalized reserve prices $r$ be $\beta$-approximate as in Definition 3.1. Fix an autobidder $i \in [K]$ who adopts a bid multiplier $\alpha_i > 1$ (see Section 2.2), so $b_i = \alpha_i v_i$. Recall $\mathcal{F}_{-i}(\cdot)$ as defined in Definition 2.4. Then the individual welfare guarantee (as defined in Definition 2.5) is bounded as:*

$$\min_{b_{-i} \in \mathcal{F}_{-i}(\alpha_i v_i)} \frac{W_i(X(b))}{OPT_i} \geq 1 - \frac{1-\beta}{\alpha_i - 1} \cdot \frac{OPT_{-i}}{OPT_i}$$
$$\text{where } OPT_{-i} = \sum_{j \neq i} OPT_j .$$

Details on implementation of VCG with personalized reserve prices can be found in Definition 2.3 and Example B.2. We defer our proof for Theorem 4.1 to Section C.1, and here we provide some intuition for the individual welfare bound in the theorem.

Given our understanding of the individual welfare metric from Definition 2.5, Theorem 4.1 implies that for a given bid multiplier $\alpha_i$, and across all potential outcomes driven by any competing bid profiles (including non-uniform bidding) that meet every bidder's ROAS constraints, bidder $i$ is guaranteed to retain at the very least a fraction of

$$1 - \frac{1-\beta}{\alpha_i - 1} \cdot \frac{OPT_{-i}}{OPT_i}$$

of the welfare anticipated in the most efficient outcome.

This welfare assurance is particularly broad in scope. It doesn't hinge on predefined assumptions about the strategic decisions of other bidders and remains applicable across all bid profiles that adhere to the bidders' ROAS criteria. An important nuance is that our outlined bounds don't rely on the notion of an autobidding equilibrium. This is rooted in the observation that in real-world applications, autobidders might not consistently reach an equilibrium state (meaning a mutually best response among bid profiles). However, typical bidding algorithms tend to stabilize on bid profiles where every bidder's constraints are met [8, 14, 16, 22]. Consequently, even if the strategies of bidder $i$'s competitors are multifaceted and tailored to achieve varying objectives, Theorem 4.1 remains applicable, contingent only on the resulting bid profile being feasible.

The main takeaway from Theorem 4.1 is that with more accurate ML-based predictions for valuations (i.e., a larger $\beta$), auctions can set higher approximate reserves, leading to enhanced individual welfare guarantees for each bidder. To provide some insight into the term $\frac{1-\beta}{\alpha_i-1} \cdot \frac{\text{OPT}_{-i}}{\text{OPT}_i}$, in the bound: Increasing either $\beta$ (through enhanced accuracy of ML-based valuations) or the bid multiplier $\alpha_i$ makes it costlier for competitors to outbid bidder $i$ in certain auctions. This makes it more challenging for them to cover the costs arising from aggressive overbidding, thereby curbing their manipulative influence and consequently bolstering the welfare guarantees for bidder $i$. This observation is consistent with the insights derived from Example 3.1. Additionally, the ratio $\frac{\text{OPT}_i}{\text{OPT}_{-i}}$ reflects bidder $i$'s relative market presence compared to other competitors. Our results indicate that a minor market presence can render bidders more susceptible to the manipulative tactics of others, leading to weaker individual welfare guarantees.

We recognize that the individual welfare lower bound guarantee in Theorem 4.1, specifically $1 - \frac{1-\beta}{\alpha_i-1} \cdot \frac{\text{OPT}_{-i}}{\text{OPT}_i}$, may be negative, rendering it meaningless for a small advertiser, specifically advertiser $i$ with a very small market share $\frac{\text{OPT}_i}{\text{OPT}_{-i}}$ resulting from a large number of bidders $N$. However, based on the proof provided for Theorem 4.1, in Section 4.1, we present a tighter individual welfare guarantee that replaces $\text{OPT}_{-i}$ in the numerator—total welfare summed over the $N - 1$ competitors of bidder $i$—with the total welfare of a potentially much smaller subset of bidder $i$'s competitors, leading to a tighter bound for small advertisers.

The following theorem states the welfare bound in Theorem 4.1 is tight; see Appendix C.3 for proof.

**Theorem 4.2 (Matching lower bound).** *For any $\beta \in (0, 1)$, $\alpha > 1$, and $R \geq \frac{1-\beta}{\alpha-1}$, there exists values $v \in \mathbb{R}_+^{N \times M}$ and $\beta$-approximate reserves $r \in \mathbb{R}_+^{N \times M}$, such that there is a bidder $i$ with multiplier $\alpha_i = \alpha$ and relative market share $\frac{\text{OPT}_i}{\text{OPT}_{-i}} = R$, who has an individual welfare guarantee*

$$\min_{b_{-i} \in \mathcal{F}_{-i}(\alpha_i v_i)} \frac{W_i(\mathcal{X}(b))}{\text{OPT}_i} = 1 - \frac{1-\beta}{\alpha_i - 1} \cdot \frac{\text{OPT}_{-i}}{\text{OPT}_i}.$$

## 4.1 A tighter individual welfare guarantee

The individual welfare lower bound in Theorem 4.1, namely $1 - \frac{1-\beta}{\alpha_i-1} \cdot \frac{\text{OPT}_{-i}}{\text{OPT}_i}$, may become small when $\text{OPT}_{-i}$ is large under a large number of bidder $N$. In this subsection, we present a tighter individual welfare guarantee that does not depend on the total efficient welfare of all bidder $i$'s competitors $\text{OPT}_{-i}$, but instead only the total welfare of a subset of $i$'s competitors. Our tighter lower bound relies on the following definitions:

$$\mathcal{L}_i = \left\{ j \in [M] : \text{OPT}_{i,j} > 0 \right\},$$
$$\mathcal{B}_i(k) = \left\{ j \in \mathcal{L}_i : v_{k,j} > 0, v_{k,j} \leq v_{i,j} \right\}, \tag{4}$$

See the definition of a maximal set cover in Eq. (8) in the Appendix C.1. Here, recall $\text{OPT}_{i,j}$ is the welfare of bidder $i$ in auction $\mathcal{A}_j$ under the efficient outcome; see Eq. (3); $\mathcal{L}_i$ is the collection of auctions wherein bidder $i$ can potentially experience a welfare loss due to competitors' overbidding; $\mathcal{B}_i(k)$ is the collection of auctions where competitor $k$ could potentially cause bidder $i$ to lose welfare (notice that competitor $k$ can win an auction $j$ only if $v_{k,j} > 0$). We

give an example of these definitions in the following: consider an instance consisting of 2 single slot VCG auctions (i.e. SPA) and 3 bidders with the following advertiser values

|          | SPA 1         | SPA 2         | SPA 3         |
|----------|---------------|---------------|---------------|
| bidder 1 | $v_{1,1} = 2$ | $v_{1,2} = 5$ | $v_{1,3} = 0$ |
| bidder 2 | $v_{2,1} = 1$ | $v_{2,2} = 1$ | $v_{2,3} = 10$ |
| bidder 3 | $v_{3,1} = 0$ | $v_{3,2} = 4$ | $v_{3,3} = 10$ |

Under the efficient outcome, bidder 1 wins auctions 1 and 2, and therefore bidder 1 can potentially lose welfare in auctions 1 and 2, so $\mathcal{L}_1 = \{1, 2\}$. Now, for competitor $k = 2$, it may be possible that bidder 2 solely outbids bidder 1 to win both auctions 1 and 2, i.e., $\mathcal{B}_1(2) = \{1, 2\}$. Similarly, for competitor $k = 3$, she can only outbid bidder 1 in auction 2 so that $\mathcal{B}_1(3) = \{2\}$.

With these definitions, we now present a tighter individual welfare guarantee than that of Theorem 4.1.

**Theorem 4.3.** *Consider that $(\mathcal{A}_j)_{j \in [M]}$ are VCG auctions, and the personalized reserve prices $r$ are $\beta$-approximate as per Definition 3.1. Fix an autobidder $i \in [K]$ who adopts a bid multiplier $\alpha_i > 1$, ensuring $b_i = \alpha_i v_i$. The individual welfare guarantee, as described in Definition 2.5, is bounded as:*

$$\min_{b_{-i} \in \mathcal{F}_{-i}(\alpha_i v_i)} \frac{W_i(\mathcal{X}(b))}{\text{OPT}_i}$$
$$\geq 1 - \frac{1-\beta}{\alpha_i - \beta} \cdot \frac{\sum_{k \in [N]/\{i\}: \mathcal{B}_i(k) \neq \emptyset} \text{OPT}_k}{\text{OPT}_i},$$

*where $\mathcal{F}_{-i}(\cdot)$ is defined in Definition 2.4.*

To illustrate the tighter welfare guarantee in Theorem 4.3 w.r.t. that in Theorem 4.1, consider the following example with $N = 3$ bidders, $M = 3$ single slot SPA auctions, and parameters $\epsilon > 0$ to be small and $X > 0$ to be large.

|          | SPA 1               | SPA 2               | SPA 3         |
|----------|---------------------|---------------------|---------------|
| bidder 1 | $v_{1,1} = 1$       | $v_{1,2} = 1$       | $v_{1,3} = 0$ |
| bidder 2 | $v_{2,1} = 1 + \epsilon$ | $v_{2,2} = 1 - \epsilon$ | $v_{2,3} = 0$ |
| bidder 3 | $v_{3,1} = 0$       | $v_{3,2} = 0$       | $v_{3,3} = X$ |

It is easy to see that when $i = 1$, $\text{OPT}_{-i} = X + 1 + \epsilon$. The set $\mathcal{L}_1$ contains auctions 1 and 2, since bidder 1 can potentially lose welfare only in these two auctions. On the other hand, we have $\mathcal{B}_1(2) = \{1, 2\}$ whereas $\mathcal{B}_1(3) = \emptyset$, so $\sum_{k \in [N]/\{i\}: \mathcal{B}_i(k) \neq \emptyset} \text{OPT}_k = v_{2,1} = 1 + \epsilon \ll \text{OPT}_{-1}$. From this example, we see that bidder $i = 1$'s welfare loss can only be driven by bidders who have direct competition with $i$, and hence bidder $i$'s welfare guarantee should only depend on the manipulative power these direct competitors have. More generally speaking, when there is less direct competition (e.g., when auctions $M$ is considerably large and each advertiser only participates in a limited amount of auctions so that the value matrix of bidders is sparse), Theorem 4.3 offers a welfare guarantee independent of $N$. Instead, it relies on a select group of competing advertisers. In keyword search advertising, the number of distinct auctions $M$ aligns with the universe of keywords queried by users on an ad platform. This number can be extremely large compared

to the number of keywords advertisers actually target, thereby making our performance guarantee in Theorem 4.3 superior to that in Theorem 4.1

## 5 VCG YIELDS BEST GUARANTEE IN BROAD CLASS OF AUCTIONS

Having presented an individual welfare guarantee in the previous Section 4 that improves based on the platform's ML advice accuracy, a natural question arises: For a given level of accuracy $\beta$, can one achieve a universally better individual welfare guarantee than that of Theorem 4.1 by considering auction formats other than VCG? In this section, we demonstrate that the answer is negative when we restrict the auction to a broad class of truthful mechanisms (Definition 2.1) and anonymous allocations (Definition 2.2).

In the subsequent theorem, we demonstrate that no allocation-anonymous, truthful auction $\mathcal{A}$ augmented by $\beta$-approximate reserves (see Definition 3.1) can universally surpass VCG. That is, for any $\mathcal{A}$, there exists a problem instance where a bidder's welfare guarantee does not exceed the individual welfare lower bound for VCG as presented in Theorem 4.1.

THEOREM 5.1. *Let $\mathcal{A}$ be any single-slot auction format (with position bias $\mu = 1$) that is allocation-anonymous, truthful, and possibly randomized. Then, there exists an instance of $M$ parallel auctions $(\mathcal{A}_j)_{j \in [M]}$ of format $\mathcal{A}$, $N$ bidders with values $v \in \mathbb{R}_+^{N \times M}$, $\beta$-approximate reserves $r \in \mathbb{R}_+^{N \times M}$, and an autobidder $i$ with multiplier $\alpha_i > 1$ (refer to Section 2.2), such that there exists a feasible bid profile $b \in \mathcal{F}$ in which $b_i = \alpha_i v_i \in \mathbb{R}_+^M$ leading to the following welfare upper bound for autobidder $i$:*

$$\frac{\mathbb{E}\left[W_i(\mathcal{X}(b))\right]}{\mathbb{E}\left[OPT_i\right]} \leq 1 - \frac{1-\beta}{\alpha_i - 1} \cdot \frac{\mathbb{E}\left[OPT_{-i}\right]}{\mathbb{E}\left[OPT_i\right]}$$

*Here, the expectation is taken with respect to possible randomness in $\mathcal{A}$.*

Our proof strategy for Theorem 5.1 hinges on the construction of a challenging autobidding instance for any given auction $\mathcal{A}$. In this instance, a specific bidder consistently finds himself with a diminished individual welfare. We showcase that, within this context, there's a bidder $i$ whose welfare is constricted to the upper bound mentioned in the theorem.

This method of construction is influenced by Example 3.1. In this example, the reduced welfare for the bidder $i$ stems from other bidders consistently outbidding $i$ in auctions where $i$'s value is the highest. These competitors then compensate for their aggressive bids using the gains from other auctions where they face little to no competition.

Building on this insight, in the problematic instance outlined in the proof of Theorem 5.1, we elevate the individual welfare for our target bidder $i$. We do this by setting up auctions where each of $i$'s competitors is the sole participant with a non-zero bid. Such "no-competition" auctions act as a buffer, allowing these competitors to recover costs associated with outbidding bidder $i$ in auctions where $i$ has the highest value. The detailed proof is available in Appendix D.2.

## 6 EXTENSION: INDIVIDUAL WELFARE GUARANTEES FOR GSP AND GFP

In this section, we extend our individual welfare guarantees for the VCG auction from Theorem 4.1 to both the GSP and GFP auctions, which are non-truthful.

Further, as discussed in Section 2.2, uniform bidding (i.e., setting the same bid multiplier for all auctions) is optimal only in truthful auctions. For GSP and GFP, instances can be constructed where non-uniform bidding strictly outperforms uniform bidding (for more details, see e.g., [17]). Therefore, for GSP and GFP autobidding instances, we do not impose any constraints on the bid values of bidders, except that they should be undominated. We define a bid value $b_i \in \mathbb{R}_+^M$ as undominated for bidder $i$ if no other bid value $b_i' \in \mathbb{R}_+^M$ exists that consistently yields a higher welfare than $b_i$ across all possible competing bid profiles. Formally, $\nexists b_i' \in \mathbb{R}_+^M$ such that $W_i(\mathcal{X}(b_i, b_{-i})) < W_i(\mathcal{X}(b_i', b_{-i}))$ for all $b_{-i} \in \mathbb{R}_+^{(N-1) \times (M)}$. The following lemma provides a lower bound for undominated bids in scenarios with $\beta$-approximate reserves.

LEMMA 6.1 (Lemma 4.7 & 4.9 of [3]). *Consider the setting where $(\mathcal{A}_j)_{j \in [M]}$ are all GSP auctions or GFP auctions, and reserve prices $r$ are $\beta$-approximate. Denote $\mathcal{U} \subseteq \mathcal{F}$ to be the set of bid profiles in which each bid is undominated and satisfies all bidders' ROAS constraints. Then for any $b \in \mathcal{U}$, $b_{i,j}$ must satisfy $b_{i,j} \geq r_{i,j} \geq \beta v_{i,j}$ for any bidder $i \in [N]$ and auction $\mathcal{A}_j$.*

Finally, our main theorem for this section is the following:

THEOREM 6.2 (INDIVIDUAL WELFARE GUARANTEE FOR GSP/GFP). *Consider that all auctions $(\mathcal{A}_j)_{j \in [M]}$ are either GSP or GFP auctions. Let the reserve prices $r$ be $\beta$-approximate, and the values $v$ be $\Delta$-separated with $\beta > \frac{\Delta}{2\Delta - 1}$. Values are termed as $\Delta$-separated if, in all auctions, the value of any bidder is at least $\Delta$ times any lesser value from a competing bidder (for a formal definition, see Definition C.2). Now, consider any undominated bid profile $b \in \mathcal{U}$, where $\mathcal{U}$ is a subset of $\mathcal{F}$ representing all undominated bids that satisfy every bidder's ROAS constraint (refer to Equation (2)). In this context, the individual welfare guarantee, as described in Definition 2.5, is bounded by:*

$$\min_{b \in \mathcal{U}} \frac{W_i(\mathcal{X}(b))}{OPT_i} \geq 1 - \frac{1-\beta}{\beta - \frac{\Delta}{2\Delta-1}} \cdot \frac{OPT_{-i}}{OPT_i}.$$

Details regarding the implementation of GSP and GFP with personalized reserve prices are provided in Definition 2.3 and further illustrated in Example B.2. The proof of Theorem 6.2 is detailed in Appendix E.1.

When we compare the individual welfare guarantees of Theorem 4.1 (for VCG) with Theorem 6.2 (for GSP/GFP), some observations emerge. Given that values are $\Delta$-separated and ML advice possesses $\beta$-accuracy, GSP/GFP offers superior individual welfare guarantees over VCG when bidders opt for smaller uniform multipliers in VCG, i.e., $\alpha_i - 1 < \beta - \frac{\Delta}{2\Delta-1}$. However, for larger multipliers, where $\alpha_i - 1 > \beta - \frac{\Delta}{2\Delta-1}$, the individual welfare within VCG surpasses that in the discussed non-truthful auctions.

## 7 NUMERICAL STUDY

Recall from Sections 4 and 6 that we theoretically demonstrated how setting personalized reserve prices with ML advice (refer to

Definition 3.1) ensures individual bidder-level worst-case welfare guarantees in VCG, GSP, and GFP auctions. Moreover, these guarantees improve as the accuracy of the ML advice increases. In this section, we provide a practical counterpart to this theoretical finding for worst-case scenarios. We illustrate how employing personalized reserve prices, informed by ML advice, can protect individual welfare on an average basis. This demonstration uses semi-synthetic data, which is derived from genuine auction data from a search ad platform.

It's worth noting that our experiments in this section are limited to VCG auctions. This choice is deliberate, primarily because bidders' bidding strategies in VCG can be confined to uniform bidding (refer to Proposition B.1 and subsequent discussions). Incorporating near-optimal or best response bidding strategies in GSP and GFP auctions adds a layer of complexity without necessarily providing new insights, as discussed in [2, 3, 16].

We obtain semi-synthetic data from ad auction logs of a search ad platform, using it to simulate VCG auctions for autobidders. Specifically, each dataset entry $j$ includes all necessary details to replicate an auction. This includes the number of ad slots sold $L_j$, the CTRs for each slot denoted as $\boldsymbol{\mu}_j = (\mu_j(1), \ldots, \mu_j(L_j)) \in [0, 1]^{L_j}$, and a list of candidate advertisers along with their respective conversion values $v_{i,j}$ and so forth. The conversion value for an advertiser represents the anticipated monetary benefit they would obtain from a user's conversion action, such as a download, email sign-up, in-app purchase, and similar actions.

It's crucial to note that our empirical study primarily aims to corroborate the central insights of our theoretical observations, rather than assessing the individual welfare of real systems in active use. For a specified accuracy level $\beta$ belonging to $\{0.25, 0.5, 0.75\}$, we independently sample $s_{i,j}^{\beta} \sim \text{Uniform}[\beta, 1]$ for each $i, j$. Subsequently, we determine the ML advice as $\underline{v}_{i,j}^{\beta} = s_{i,j}^{\beta} \cdot v_{i,j}$. Clearly, the ML advice generated in this manner is $\beta$-accurate, meaning $\underline{v}_{i,j}^{\beta}$ lies in the interval $[\beta v_{i,j}, v_{i,j}]$, consistent with Definition 3.1.

**Calculating uniform bid multipliers.** For simplicity, and with a slight departure from standard notation, we let the accuracy level $\beta = 0$ signify the imposition of no-reserve prices. Using a reserved portion of our data, we determine each advertiser's uniform bid multiplier for every accuracy level $\beta$ in the set $\{0.25, 0.5, 0.75\}$. We employ gradient descent for this task, mimicking prevalent uniform bidding practices. This choice is motivated by the widespread use of descent/primal-dual methods in real-world autobidding. These techniques have demonstrated near-optimal convergence and commendable performance guarantees, as evidenced by works such as [1, 4, 39, 45]. After these computations, we obtain our bid multipliers denoted $\alpha_i^{\beta}$. For a comprehensive explanation on calculating these uniform bid multipliers, we direct readers to Appendix F.1.

For each accuracy level $\beta$ in the set $\{0, 0.25, 0.5, 0.75\}$ (where $\beta = 0$ represents the control experiment without personalized reserve prices), let $W_i^{\beta}$ denote the realized total welfare for advertiser $i \in [N]$ across $M$ VCG auctions. This welfare is calculated with respect to bid multipliers $(\alpha_i^{\beta})_{i \in [N]}$, values $\boldsymbol{v} = (v_{i,j})_{i \in [N], j \in [M]}$, ad slot counts $(L_j)_{j \in [M]}$, and CTRs $(\boldsymbol{\mu}_j)_{j \in [M]}$.

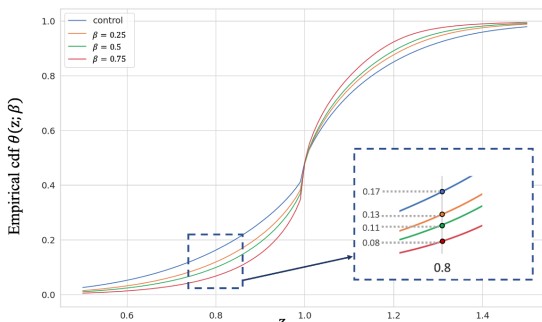

**Figure 1: Empirical cumulative distribution function (CDF)** $\theta(z; \beta) = \frac{1}{N} \sum_{i \in [N]} \mathbb{I}\left(\frac{W_i^{\beta}}{\text{OPT}_i} \leq z\right)$ **for various accuracy levels:** $\beta = 0$ **(no personalized reserve prices),** $0.25, 0.5,$ **and** $0.75$. **For a given** $z < 1$, **the empirical CDF decreases as** $\beta$ **increases, reflecting improved accuracy. For instance, at** $z = 0.8$, **we have** $\theta(0.8; 0) = 0.17$, $\theta(0.8; 0.25) = 0.13$, $\theta(0.8; 0.5) = 0.11$, **and** $\theta(0.8; 0.75) = 0.08$.

Given that $\text{OPT}_i$ represents the welfare of bidder $i \in [N]$ under the efficient outcome (as defined in Eq. (3)), we introduce the empirical cumulative distribution function (CDF) $\theta(z; \beta)$ to represent the proportion of advertisers whose realized welfare does not exceed $z$ times their efficient outcome welfare. Here, $z$ can be either less or greater than 1. Specifically, $\theta(z; \beta)$ for $z < 1$ shows the proportion of advertisers who experience a welfare deficit relative to $\text{OPT}_i$.

In Figure 1, we depict $\theta(z; \beta)$ for the given $\beta$ values. Observations show that, for $z < 1$, $\theta(z; \beta)$ diminishes as $\beta$ rises. This indicates that enhancing the accuracy of ML advice in setting personalized reserve prices reduces the proportion of advertisers enduring a welfare reduction relative to the efficient outcome. For instance, when $z = 0.8$, indicating a 20% welfare loss from the efficient outcome, the proportion of advertisers facing this loss is 17% without reserve prices ($\beta = 0$). This proportion reduces to 13%, 11%, and 8% for $\beta$ values of 0.25, 0.5, and 0.75, respectively. The primary insight is that better ML advice empowers the ad platform to reduce the proportion of advertisers suffering from welfare losses compared to the efficient outcome. Another observation is that the empirical CDF tends toward a step function with a significant shift at $z = 1$, suggesting that as ML accuracy improves, an increasing number of advertisers approach their efficient outcome welfare.

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

Appendices for

# Individual Welfare Guarantees in the Autobidding World with Machine-learned Advice

## A  EXTENDED LITERATURE REVIEW

**Mechanism design with constrained bidders.** This work considers autobidders who aim to maximize welfare while respecting an ROAS constraint that ensures total spend is less than total acquired value, and thus our work relates to the general theme of mechanism design in the presence of constrained agents. The works of [29, 41] study revenue-optimal auction design when bidders who maximize quasi-linear utility are constrained by budgets, and return-on-investment (ROI), respectively. [7] study revenue-maximizing auctions for ROI constrained bidders under different objectives and information structures for values and ROI targets. This work differs from these papers as we do not study new auction formats and platform revenue-optimization, but instead presents insights into how to incorporate ML advice as reserves in classic auctions like VCG, GSP and GFP can improve individual bidder welfare.

**Algorithmic bidding/learning under constraints** The behavior of our bidders of interest are governed by their ROAS constraints, and there has been a growing area of works on bidding-algorithm design under similar financial constraints for online advertising markets. [4] develops theoretical performance guarantees of the budget pacing strategy for bidders with hard budget cap (see more on budget management strategies in [5]), while [6] presents a more general mirror descent algorithm for online resource allocation problems. [25] present near-optimal bidding algorithms for bidders with both budget and ROI constraints in expectation. Finally, [13] study a multi-channel ad procurement problem under the autobidding setup where ad platforms, i.e. channels, autobid on behalf of advertisers, and the work develops algorithms that optimizes advertisers' interaction decisions with channels to maximize conversion. In this work, we do not study the design of bidding algorithms but instead consider worst case outcomes under any feasible bidding profile.

**Reserve price optimization.** Reserve price techniques and optimization have been studied for different auction formats and settings. In the single-shot second price auction setting [9, 18, 40] presents different approaches with theoretical performance guarantees to optimize personalized reserve prices, while [49] presents an empirical study on the impact of reserve price on the entire auction system for display advertising. For repeated second price auctions, [24, 28, 32] dynamically learn reserve prices to maximize cumulative revenue facing strategic agents, where as [20] optimize reserve prices to balance revenue and bidders' incentives to misreport. For first price auctions, [21] introduces a gradient-based adaptive algorithm to dynamically optimize reserve prices. Nevertheless, all aforementioned works attempt to design and learn optimal or near optimal reserve prices for the purpose of revenue maximization, whereas in our work we directly set reserves using ML advice provided by some external black-box, and shed light on how reserve prices can improve individual welfare among all bidders.

## B  ADDITIONAL MATERIALS FOR SECTION 2

**Example B.1** (Example for allocation anonymous auctions). *Consider a single GSP auction with 2 slots and 3 bidders who submitted a bid profile $\boldsymbol{b} = (0.1, 0.2, 0.3)$. As GSP allocates slots by ranking bidders' submitted bids, the outcome under bid profile $\boldsymbol{b}$ is $\boldsymbol{x} = \begin{pmatrix} 0,0 \\ 0,1 \\ 1,0 \end{pmatrix}$. Next, consider some permutation $\sigma$ that maps $\{1,2,3\}$ to $\{3,1,2\}$. That is, $\sigma(1) = 3$, $\sigma(2) = 1$ and $\sigma(3) = 2$. Under this permutation, the corresponding permuted bid profile $\boldsymbol{b}' = (0.3, 0.1, 0.2)$, which results in the outcome $\boldsymbol{x}' = \begin{pmatrix} 1,0 \\ 0,0 \\ 0,1 \end{pmatrix}$. Then, it is easy to check that $\mathbb{P}(x_{\sigma(i),j} = 1) = \mathbb{P}(x'_{i,j} = 1) = \begin{cases} 1 & \text{if } (i,j) = (1,1) \text{ or } (3,2) \\ 0 & \text{otherwise} \end{cases}$. In particular, because $\sigma(1) = 3$ we have $\mathbb{P}(x_{3,1} = 1) = \mathbb{P}(x'_{1,1} = 1) = 1$, and because $\sigma(3) = 2$ we have $\mathbb{P}(x_{2,2} = 1) = \mathbb{P}(x'_{3,2} = 1) = 1$.*

**Example B.2** (Personalized-reserve augmented VCG, GSP, GFP auctions). *Consider $M \geq 2$ parallel position auctions $(\mathcal{A}_j)_{j \in [M]}$ all of which take the form of VCG, GSP or GFP auctions. Each auction $\mathcal{A}_j$ is associated with $L_j \geq 1$ slots and CTRs $\boldsymbol{\mu}_j = (\mu_j(\ell))_{\ell \in L_j}$. Assume $N$ bidders submit bid profile $\boldsymbol{b}_j \in \mathbb{R}_+^N$ to auction $\mathcal{A}_j$, where $\widetilde{N}_j \leq N$ are cleared, i.e. greater than respective personalized reserve prices. Define $\widetilde{\boldsymbol{b}}_j \in \mathbb{R}_+^{\widetilde{N}_j}$ to be all "cleared bids", and let $\widetilde{b}_j^{(\ell)}$ be the $\ell$th highest cleared bid. Then, in $\mathcal{A}_j$ bidders who cleared their reserves are assigned slots according to the ranking of $\widetilde{\boldsymbol{b}}_j$, whereas the bidders who do not clear their reserves never get allocated any slots. The payment for a bidder $i$ who cleared her reserve and allocated slot $\ell_{i,j} \in [\min\{\widetilde{N}_j, L_j\}]$ is*

- *VCG: $p_{i,j} = \sum_{\ell=\ell_{i,j}}^{\min\{\widetilde{N}_j, L_j\}} (\mu_j(\ell) - \mu_j(\ell+1)) \cdot \max\{\widetilde{b}_j^{(\ell+1)}, r_{i,j}\}$ where $\widetilde{b}_j^{(\ell)} = 0$ when $\ell > \widetilde{N}_j$.*
- *GSP: $p_{i,j} = \mu_j(\ell_{i,j}) \cdot \max\{\widetilde{b}_j^{(\ell_{i,j}+1)}, r_{i,j}\}$.*

- *GFP:* $p_{i,j} = \mu_j(\ell_{i,j}) \cdot \max\{\widetilde{b}_j^{(\ell_{i,j})}, r_{i,j}\}$.

It is well known that for the same bid profile $\boldsymbol{b}$ and for any bidder $i$, the payment under the GFP auction is greater than equal to that under GSP auction, and the payment under the GSP auction is greater than equal to that under VCG; see e.g. [19].

*Uniform bidding.* The following proposition shows uniform bidding is the optimal bidding strategy in truthful auctions.

**Proposition B.1** (Uniform bidding is optimal for autobidders in truthful auctions). *Let all auctions $(\mathcal{A}_j)_{j\in[M]}$ be identical truthful auctions (see Definition 2.1), and bidder $i \in [N]$ is an autobidder who aims to maximize welfare $\mathbb{E}[W_i(\mathcal{X}(\boldsymbol{b}_i, \boldsymbol{b}_{-i}))]$ subject to the ROAS constraint in Eq. (2) for any fixed competing bids $\boldsymbol{b}_{-i} \in \mathbb{R}_+^{N-1}$. Then, there exists some constant uniform multiplier $\alpha_i^* \geq 1$ s.t. the uniform bidding profile $\alpha_i^* \boldsymbol{v}_i$ is $i$'s optimal strategy:*

$$\alpha_i^* \cdot \boldsymbol{v}_i \in \arg\max_{\boldsymbol{b}_i \in \mathbb{R}_+^M} \mathbb{E}[W_i(\mathcal{X}(\boldsymbol{b}_i, \boldsymbol{b}_{-i}))] \quad s.t. \quad \mathbb{E}[W_i(\mathcal{X}(\boldsymbol{b}_i, \boldsymbol{b}_{-i}))] \geq \mathbb{E}[\mathcal{P}_i(\boldsymbol{b}_i, \boldsymbol{b}_{-i})], . \tag{5}$$

*Further, adopting any uniform bid multiplier $\alpha_i < 1$ is weakly dominated by truthful bidding, i.e. $\mathbb{E}[W_i(\mathcal{X}(\alpha_i \boldsymbol{v}_i, \boldsymbol{b}_{-i}))] \leq \mathbb{E}[W_i(\mathcal{X}(\boldsymbol{v}_i, \boldsymbol{b}_{-i}))]$ for any $\boldsymbol{b}_{-i} \in \mathbb{R}_+^{N-1}$.*

This is a well-known result that has been proved and adopted in many related works such as [1, 3, 16, 37] and we will omit the proof here.

# C   ADDITIONAL MATERIALS FOR SECTION 4

## C.1   Proof for Theorem 4.1

First, to prove Theorem 4.1, we rely on the definition of an advertisers' loss in welfare compared to her welfare contribution under the efficient outcome, formally defined as followed:

**Definition C.1** (Welfare loss w.r.t. efficient outcome). *For any bidder $i \in [N]$ and outcome $\boldsymbol{x} = (\boldsymbol{x}_j \in \{0,1\}^{N \times L_j})_{j\in[M]}$, let $\mathcal{L}_i(\boldsymbol{x}) = \{j \in [M] : W_{i,j}(\boldsymbol{x}) < OPT_{i,j}\}$ be the set of auctions in which bidder $i$'s acquired welfare is less than that of her welfare under the efficient outcome. Then, we define the welfare loss of bidder $i$ under outcome $\boldsymbol{x}$ w.r.t. the efficient outcome $\boldsymbol{x}^*$ as:*

$$LOSS_i(\boldsymbol{x}) = \sum_{j \in \mathcal{L}_i(\boldsymbol{x})} (OPT_{i,j} - W_{i,j}(\boldsymbol{x})) . \tag{6}$$

**Remark C.1.** *For any outcome $\boldsymbol{x}$, let $\ell_{i,j}$ be the position (i.e. ranking) of bidder $i$ in auction $j$, and recall that $\ell_{i,j}^*$ is the position of bidder $i$ in auction $j$ under the efficient outcome $\boldsymbol{x}^*$. Then, the set $\mathcal{L}_i(\boldsymbol{x}) = \{j \in [M] : W_{i,j}(\boldsymbol{x}_j) < OPT_{i,j}\}$ (where $W_{i,j}(\boldsymbol{x}_j)$ is bidder $i$'s welfare in $\mathcal{A}_j$ as defined in Eq.(1)) can also be interpreted as the set of auctions where bidder $i$'s ranking under $\boldsymbol{x}$ is lower than her ranking under $\boldsymbol{x}^*$, or in other words the set of auctions that incur a welfare loss w.r.t. $\boldsymbol{x}^*$. Hence we can also rewrite $\mathcal{L}_i(\boldsymbol{x}) = \{j \in [M] : \ell_{i,j} > \ell_{i,j}^*\}$.*

The following proposition connects the notion of welfare loss (as in Definition C.1) and individual welfare (as in Definition 2.5) by showing an upper bound on welfare loss can be directly translated into a welfare lower bound that corresponds to our individual welfare guarantee.

**Proposition C.1** (Translating loss to individual welfare guarantee). *Assume for bidder $i \in [N]$ and outcome $\boldsymbol{x} = (\boldsymbol{x}_j \in \{0,1\}^{N \times L_j})_{j\in[M]}$ we have $LOSS_i(\boldsymbol{x}) \leq B$ for some $B > 0$. Then, $\frac{W_i(\boldsymbol{x})}{OPT_i} \geq 1 - \frac{B}{OPT_i}$.*

The proof of this proposition is presented in Section C.4. Now, in light of this proposition, we proceed to prove Theorem 4.1 by bounding bidder $i$'s welfare loss for auctions where she obtains a slot that is lower in position than what she would have obtained under the efficient outcome.

PROOF OF THEOREM 4.1. Fix any feasible competing bid profile $\boldsymbol{b}_{-i} \in \mathcal{F}_{-i}(\alpha_i \boldsymbol{v}_i)$ under which every bidders' ROAS constraint is satisfied; see Definition 2.4. Denote the corresponding outcome as $\boldsymbol{x} = \mathcal{X}(\boldsymbol{b})$, and $\ell_{k,j}, \ell_{k,j}^*$ to be the position of any bidder $k \in [N]$ in auction $j \in [M]$ under outcome $\boldsymbol{x}$ and the efficient outcome, respectively.

Consider any auction $j \in \mathcal{L}_i(\boldsymbol{x}) = \{j \in [M] : \ell_{i,j} > \ell_{i,j}^*\}$ (see Remark C.1), i.e. in auction $\mathcal{A}_j$, bidder $i$ acquires a position (under $\boldsymbol{x}$) bellow her position in the efficient outcome $\boldsymbol{x}^*$. This implies there must exist competing bidders in auction $\mathcal{A}_j$ whose values are smaller than that of bidder $i$'s, but obtains a better position, making bidder $i$ lose welfare. Motivated by this, we let $\mathcal{B}_i(k; \boldsymbol{x})$ denote the set of all auctions in which bidder $k$'s value is lower than $i$'s but acquires a better position than $i$:

$$\mathcal{B}_i(k; \boldsymbol{x}) = \left\{ j \in [M] : OPT_{i,j} > 0, \; v_{k,j} < v_{i,j} \text{ and } \ell_{k,j} \leq \ell_{i,j}^* < \ell_{i,j} \right\} \tag{7}$$

where we recall $OPT_{i,j}$ is the welfare of bidder $i$ in auction $j$ under the efficient outcome. Further, we can find a collection of $i$'s competitors whose $\mathcal{B}_i(\,\cdot\,; \boldsymbol{x})$ "covers" all auctions $\mathcal{L}_i(\boldsymbol{x})$ in which $i$ loses welfare. We call this collection of competitors a covering, and formally define the collection of all coverings, called $C_i(\boldsymbol{x})$, as followed:

$$C_i(\boldsymbol{x}) = \left\{ C \subseteq [N]/\{i\} : (\mathcal{B}_i(k; \boldsymbol{x}))_{k \in C} \text{ is a maximal set cover of } \mathcal{L}_i(\boldsymbol{x}) \right\} . \tag{8}$$

Here, for any set $\mathcal{S}$, we say $\mathcal{S}_1 \ldots \mathcal{S}_n$ a maximal set cover of $\mathcal{S}$ if $\mathcal{S} \subseteq \bigcup_{n' \in [n]} \mathcal{S}_{n'}$ but $\mathcal{S} \subsetneq \bigcup_{n' \in [n]} \mathcal{S}_{n'} / \mathcal{S}_{n''}$ for any $n'' \in [n]$. In words, $\mathcal{B}_i(k; \boldsymbol{x})$ is the set of auctions in which bidder $k$ has a smaller value than bidder $i$ but acquires a better position, and any $C \in C_i(\boldsymbol{x})$ is a subset of $i$'s competitors who are responsible for all welfare losses of bidder $i$ in auctions of $\mathcal{L}_i(\boldsymbol{x})$.

Fix any covering $C \in C_i(\boldsymbol{x})$, and some bidder $k \in C$. We first state the following inequality that bounds the welfare loss of bidder $i$ caused by competitor $k \in C$ in the covering (we will prove this inequality later).

$$\sum_{j \in \mathcal{B}_i(k; \boldsymbol{b})} \left( \mu(\ell^*_{i,j}) - \mu(\ell_{i,j}) \right) v_{i,j} \leq \frac{1 - \beta}{\alpha_i - \beta} \sum_{j \in [M]} \mu(\ell_{k,j}) v_{k,j} \tag{9}$$

Summing the above over all competitors $k \in C$, we have

$$\begin{aligned}
\text{LOSS}_i(\boldsymbol{x}) &= \sum_{j \in \mathcal{L}_i(\boldsymbol{x})} \left( \mu(\ell^*_{i,j}) - \mu(\ell_{i,j}) \right) v_{i,j} \overset{(a)}{\leq} \sum_{k \in C} \sum_{j \in \mathcal{B}_i(k; \boldsymbol{b})} \left( \mu(\ell^*_{i,j}) - \mu(\ell_{i,j}) \right) v_{i,j} \\
&\overset{(b)}{\leq} \frac{1 - \beta}{\alpha_i - \beta} \sum_{k \in C} \sum_{j \in [M]} \mu(\ell_{k,j}) v_{k,j} = \frac{1 - \beta}{\alpha_i - \beta} \sum_{k \in C} W_k(\boldsymbol{x}) \\
&\leq \frac{1 - \beta}{\alpha_i - \beta} W_{-i}(\boldsymbol{x}) \\
&\overset{(c)}{\leq} \frac{1 - \beta}{\alpha_i - \beta} \left( \text{OPT}_{-i} + \text{LOSS}_i(\boldsymbol{x}) \right) \\
\implies \text{LOSS}_i(\boldsymbol{x}) &\leq \frac{1 - \beta}{\alpha_i - 1} \text{OPT}_{-i}.
\end{aligned} \tag{10}$$

Here, in (a) we used the fact that $\mathcal{L}_i(\boldsymbol{x}) \subseteq \bigcup_{k \in C} \mathcal{B}_i(k; \boldsymbol{x})$ (see Eq. (8)); in (b) we applied Eq. (9); (c) follows from $\text{OPT} \geq \sum_{i \in [N]} W_i(\boldsymbol{x})$ where OPT is the total efficient welfare and $\sum_{i \in [N]} W_i(\boldsymbol{x})$ is the total welfare under outcome $\boldsymbol{x}$, so further

$$\begin{aligned}
\text{OPT}_{-i} &\geq W_{-i}(\boldsymbol{x}) + W_i(\boldsymbol{x}) - \text{OPT}_i \\
&= W_{-i}(\boldsymbol{x}) + \sum_{j \in \mathcal{L}_i(\boldsymbol{x})} \left( W_{i,j}(\boldsymbol{x}) - \text{OPT}_{i,j} \right) + \sum_{j \in [M] / \mathcal{L}_i(\boldsymbol{x})} \left( W_{i,j}(\boldsymbol{x}) - \text{OPT}_{i,j} \right) \\
&\overset{(e)}{\geq} W_{-i}(\boldsymbol{x}) + \sum_{j \in \mathcal{L}_i(\boldsymbol{x})} \left( W_{i,j}(\boldsymbol{x}) - \text{OPT}_{i,j} \right) \\
&= W_{-i}(\boldsymbol{x}) - \text{LOSS}_i(\boldsymbol{x}).
\end{aligned} \tag{11}$$

where in (e) we used the fact that $W_{i,j}(\boldsymbol{x}) \geq \text{OPT}_{i,j}$ in any auction $j \in [M] / \mathcal{L}_i(\boldsymbol{x})$. Finally, applying Proposition C.1 w.r.t. upper bound of $\text{LOSS}_i(\boldsymbol{x})$, and noting that the feasible competing bid profile is arbitrary, we obtain the desired welfare guarantee lower bound.

Now, it remains to prove Eq. (9) that bounds the welfare loss of bidder $i$ caused by competitor $k \in C$ in the covering. Denote $p_{k,j}$ as the payment of bidder $k$, and $\hat{b}_{\ell,j}$ as the $\ell$th largest bid in any auction $j \in [M]$. Then in some auction $j \in \mathcal{B}_i(k; \boldsymbol{b})$ recall from Eqs. (7) and (8) that $v_{k,j} < v_{i,j}$ but $\ell_{k,j} \leq \ell^*_{i,j} < \ell_{i,j}$. Thus bidder $k$'s payment is lower bounded as

$$\begin{aligned}
\text{For } j \in \mathcal{B}_i(k; \boldsymbol{b}), \quad p_{k,j} &\overset{(a)}{\geq} \sum_{\ell = \ell_{k,j}}^{L_j} \left( \mu(\ell) - \mu(\ell + 1) \right) \hat{b}_{\ell+1,j} \\
&= \sum_{\ell = \ell_{k,j}}^{\ell^*_{i,j} - 1} \left( \mu(\ell) - \mu(\ell + 1) \right) \hat{b}_{\ell+1,j} + \sum_{\ell = \ell^*_{i,j}}^{\ell_{i,j} - 1} \left( \mu(\ell) - \mu(\ell + 1) \right) \hat{b}_{\ell+1,j} + p_{i,j} \\
&\overset{(b)}{\geq} \left( \mu(\ell_{k,j}) - \mu(\ell^*_{i,j}) \right) v_{i,j} + \alpha_i \left( \mu(\ell^*_{i,j}) - \mu(\ell_{i,j}) \right) v_{i,j} + \beta \cdot \mu(\ell_{i,j}) v_{i,j} \\
&= \mu(\ell_{k,j}) v_{i,j} + (\alpha_i - 1) \left( \mu(\ell^*_{i,j}) - \mu(\ell_{i,j}) \right) v_{i,j} - (1 - \beta) \cdot \mu(\ell_{i,j}) v_{i,j}.
\end{aligned} \tag{12}$$

Here, (a) follows from the VCG payment rule (see Example B.2); (b) follows from the fact that bidder $i$'s ranking is $\ell_{i,j}$, so any bidder who is ranked before position $\ell_{i,j}$ submits a bid greater than bidder $i$'s bid $b_{i,j} = \alpha_i v_{i,j}$, i.e. $\hat{b}_{\ell,j} \geq b_{i,j} = \alpha_i v_{i,j} > v_{i,j}$ for any $\ell \leq \ell_{i,j}$.

On the other hand, we have

$$\sum_{j \in \mathcal{B}_i(k; \boldsymbol{b})} p_{k,j} + \sum_{j \notin \mathcal{B}_i(k; \boldsymbol{b})} p_{k,j} \leq \sum_{j \in \mathcal{B}_i(k; \boldsymbol{b})} \mu(\ell_{k,j}) v_{k,j} + \sum_{j \notin \mathcal{B}_i(k; \boldsymbol{b})} \mu(\ell_{k,j}) v_{k,j}$$

$$p_{k,j} \geq \beta \cdot \mu(\ell_{k,j}) v_{k,j} \quad \forall j \in [M],$$

where the first inequality follows from bidder $k$'s ROAS constraint; the second inequality follows from the fact that any winning bidder's payment must be greater than her $\beta$-approximate reserves. Combining the above inequalities and rearranging we get

$$\sum_{j \in \mathcal{B}_i(k;\boldsymbol{b})} p_{k,j} \leq \sum_{j \in \mathcal{B}_i(k;\boldsymbol{b})} \mu(\ell_{k,j}) v_{k,j} + (1-\beta) \cdot \sum_{j \notin \mathcal{B}_i(k;\boldsymbol{b})} \mu(\ell_{k,j}) v_{k,j}, \tag{13}$$

Summing Eq.(12) over all $j \in \mathcal{B}_i(k;\boldsymbol{b})$ and combining with Eq. (13), we get

$$(\alpha_i - 1) \cdot \sum_{j \in \mathcal{B}_i(k;\boldsymbol{b})} \left( \mu(\ell_{i,j}^*) - \mu(\ell_{i,j}) \right) v_{i,j}$$

$$\leq (1-\beta) \cdot \left( \sum_{j \in \mathcal{B}_i(k;\boldsymbol{b})} \mu(\ell_{i,j}) v_{i,j} + \sum_{j \notin \mathcal{B}_i(k;\boldsymbol{b})} \mu(\ell_{k,j}) v_{k,j} \right) + \sum_{j \in \mathcal{B}_i(k;\boldsymbol{b})} \mu(\ell_{k,j}) \left( v_{k,j} - v_{i,j} \right)$$

$$\overset{(a)}{\leq} (1-\beta) \cdot \left( \sum_{j \in \mathcal{B}_i(k;\boldsymbol{b})} \mu(\ell_{i,j}) v_{i,j} + \sum_{j \notin \mathcal{B}_i(k;\boldsymbol{b})} \mu(\ell_{k,j}) v_{k,j} + \sum_{j \in \mathcal{B}_i(k;\boldsymbol{b})} \mu(\ell_{k,j}) \left( v_{k,j} - v_{i,j} \right) \right)$$

$$\overset{(b)}{\leq} (1-\beta) \cdot \left( \sum_{j \in \mathcal{B}_i(k;\boldsymbol{b})} \mu(\ell_{i,j}) v_{i,j} + \sum_{j \in [M]} \mu(\ell_{k,j}) v_{k,j} - \sum_{j \in \mathcal{B}_i(k;\boldsymbol{b})} \mu(\ell_{i,j}^*) v_{i,j} \right).$$

In (a), we used the fact that $\beta \in (0,1]$ and $v_{k,j} - v_{i,j} < 0$ for any $k \in C \subseteq C_i(\boldsymbol{x})$; see definition of $C_i(\boldsymbol{x})$ in Eq. (8); and (b) follows from $\ell_{k,j} \leq \ell_{i,j}^*$ for any $k \in C \subseteq C_i(\boldsymbol{x})$. Rearranging terms we obtain the desired Eq. (9). □

## C.2 Applicability of the individual welfare guarantee when all bidders bid uniformly

We recognize that as the individual welfare lower bound in Theorem 4.1 monotonically increases in the bid multiplier $\alpha_i$, it is tempting for bidder $i$ to apply a very large multiplier $\alpha_i$. Nevertheless, in this section we describe a potential tradeoff between large multipliers (i.e. better individual welfare guarantees in light of Theorem 4.1) and ROAS feasibility in the practical scenario where all bidders are autobidders and adopt uniform bidding.

To illustrate, we see that for large multiplier $\alpha_i$, the set of competing bids $\mathcal{F}_{-i}(\alpha_i \boldsymbol{v}_i)$ may only include very small bid values (e.g. the bid profile where each competing bidder (under)bids some small $\epsilon > 0$ close to 0 in each auction), at which bidder $i$ faces nearly no competition so that the ROAS constraint can be trivially satisfied for every bidder. In light of this discussion, we consider a more practical scenario where all competing bidders are also autobidders and adopt uniform bidding, or equivalently, a refinement of $\mathcal{F}_{-i}(\alpha_i \boldsymbol{v}_i)$ in which each competing bidder $j \neq i$, similar to bidder $i$, also adopts uniform bidding with bid multiplier $\alpha_j \geq 1$. We define $\mathcal{F}_{-i}^u(\boldsymbol{b}_i) = \mathcal{F}_{-i}(\boldsymbol{b}_i) \cap \{(\alpha_j \boldsymbol{v}_j)_{j \neq i} : \alpha_j \geq 1\}$ that represents the set of uniform competing bids for bidder $i$ that ensure ROAS constraint satisfaction for every bidder. From Theorem 4.1, it is easy to see

$$\min_{\boldsymbol{b}_{-i} \in \mathcal{F}_{-i}^u(\alpha_i \boldsymbol{v}_i)} \frac{W_i(\mathcal{X}(\boldsymbol{b}))}{\text{OPT}_i} \overset{(i)}{\geq} 1 - \frac{1-\beta}{\alpha_i - 1} \cdot \frac{\text{OPT}_{-i}}{\text{OPT}_i}, \tag{14}$$

where (i) follows from $\min_{\boldsymbol{b}_{-i} \in \mathcal{F}_{-i}^u(\alpha_i \boldsymbol{v}_i)} \frac{W_i(\mathcal{X}(\boldsymbol{b}))}{\text{OPT}_i} \geq \min_{\boldsymbol{b}_{-i} \in \mathcal{F}_{-i}(\alpha_i \boldsymbol{v}_i)} \frac{W_i(\mathcal{X}(\boldsymbol{b}))}{\text{OPT}_i}$ because $\mathcal{F}_{-i}^u(\alpha_i \boldsymbol{v}_i) \subseteq \mathcal{F}_{-i}(\alpha_i \boldsymbol{v}_i)$. Nevertheless, in light of Eq. (14), when all bidders bid uniformly, an excessively large $\alpha_i$ may let bidder $i$ incur large payments that significantly exceed her values, resulting in non-existence of competing uniform bids $\boldsymbol{b}_{-i}$ that can ensure satisfaction of every bidders' ROAS constraints, i.e. $\mathcal{F}_{-i}^u(\alpha_i \boldsymbol{v}_i)$ being empty. In other words, there exists a tradeoff between large multipliers (i.e. better individual welfare guarantees) and ROAS feasibility when all bidders bid uniformly. The following Lemma C.2, along with a technical definition of "well-separated" values per Definition C.2, addresses this tradeoff by characterizing how large the multiplier $\alpha_i$ can be set that still ensures the existence of uniform competing bids within $\mathcal{F}_{-i}^u(\alpha_i \boldsymbol{v}_i)$.

**Definition C.2** ($\Delta$-separated values). *We say values $\boldsymbol{v} \in \mathbb{R}_{\geq 0}^{N \times M}$ are $\Delta$-separated for some $\Delta > 1$ if any value $v_{i,j}$ is at least $\Delta$ times as much as any value that is less than $v_{i,j}$ in the same auction $j$, i.e. $v_{i,j} \geq \Delta \cdot \max\{v_{k,j} : k \in [N], v_{k,j} < v_{i,j}\}$ for any bidder $i$ and auction $j$.*[4]

**Lemma C.2** (Valid regions for uniform bid multiplier). *Let $(\mathcal{A}_j)_{j \in [M]}$ be VCG auctions and assume bidders values are $\Delta$-separated (Definition C.2) in every auction for some $\Delta > 1$, then $\mathcal{F}_{-i}^u(\alpha_i \boldsymbol{v}_i) \neq \varnothing$ for any $\alpha_i \in [1, \Delta)$.*

PROOF. Recall there is $\Delta$-separation in values. Fix a bidder $i$ and let $v_j^+$ be the smallest competitor value that is strictly greater than $v_{i,j}$ in any auction $\mathcal{A}_j$ where bidder $i$'s value is not the largest, and by definition of $\Delta$-separated values we have $v_j^+ \geq \Delta v_{i,j}$. Hence, by using any multiplier $\alpha_i \in [1, \Delta)$ and assuming competitors bid truthfully, the outcome of the auctions would be identical to that of everyone (including bidder $i$) bidding truthfully. And since truthful bidding is always feasible, we conclude that $\boldsymbol{v}_{-i} \in \mathcal{F}_{-i}^u(\alpha_i \boldsymbol{v}_i)$ for $\alpha_i \in [1, \Delta)$. □

---

[4] Definition C.2 also captures values which are "additively separated". In particular, take some $d > 0$ such that $d < \min\{v_{i,j} : v_{i,j} \neq 0\}$ and also $v_{i,j} - d \geq \max\{v_{k,j} : k \in [N], v_{k,j} < v_{i,j}\}$ for any bidder $i$ and auction $j$. Then, by taking $\Delta \in \min_{v_{i,j} : v_{i,j} \neq 0} \left\{ \frac{v_{i,j}}{v_{i,j} - d} \right\}$, the values are $\Delta$-separated according to Definition C.2 because $\frac{1}{\Delta} v_{i,j} \geq v_{i,j} - d \geq \max\{v_{k,j} : k \in [N], v_{k,j} < v_{i,j}\}$ for all $v_{i,j}$. This suggests Definition C.2 is quite general to capture value separation scenarios.

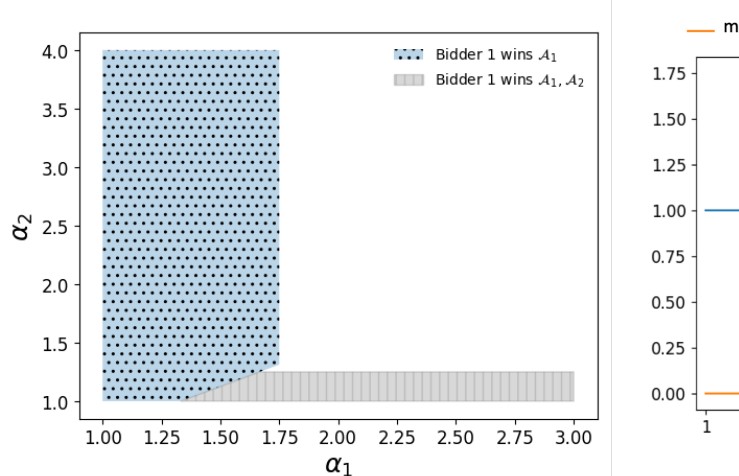 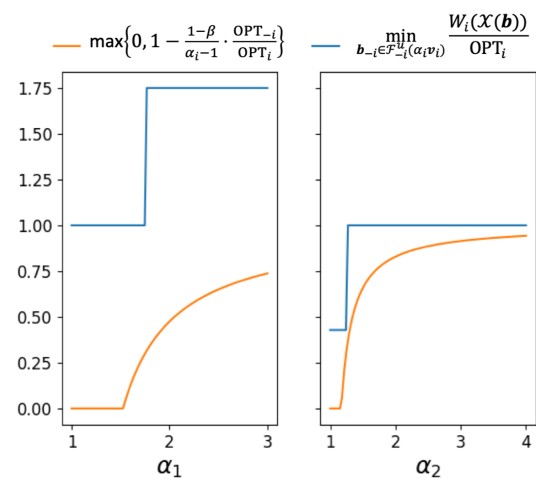

**Figure 2: Left: Two colored regions represent uniform bid multipliers** $(\alpha_1, \alpha_2) \in [1, \infty)^2$ **that lead to feasible bid profiles** $(\boldsymbol{b}_1, \boldsymbol{b}_2) \in \mathcal{F}$. **Right: Comparison between the individual welfare guarantee of Theorem 4.1, namely** $1 - \frac{1-\beta}{\alpha_i - 1} \cdot \frac{\text{OPT}_{-i}}{\text{OPT}_i}$, **and the worst case welfare for each bidder** $i$ **(normalized by** $\text{OPT}_i$**) among all feasible bid profiles when both bidders adopt uniform bidding, namely** $\min_{\boldsymbol{b}_{-i} \in \mathcal{F}^u_{-i}(\alpha_i \boldsymbol{v}_i)} \frac{W_i(\mathcal{X}(\boldsymbol{b}))}{\text{OPT}_i}$.

We also remark that the upper bound $\Delta$ in Lemma C.2 is sufficient, meaning that there may exist larger values of $\alpha_i$ that can ensure the set $\mathcal{F}^u_{-i}(\alpha_i \boldsymbol{v}_i) \neq \varnothing$ nonempty. To better visualize the structure of $\mathcal{F}^u_{-i}(\alpha_i \boldsymbol{v}_i)$, as well as our individual welfare guarantee in Theorem 4.1 and Eq. (14), we present the following example.

**Example C.1.** *Consider 2 bidders bidding in 3 single-slot VCG auctions in which each slot is associated with CTR equal to 1. Bidder values are* $\boldsymbol{v}_1 = (4, 3, 1)$ *and* $\boldsymbol{v}_2 = (1, 4, 3)$, *while personalized reserves are set to be* $\boldsymbol{r}_i = \beta \boldsymbol{v}_i$ *for* $\beta = 0.7$ *and* $i = 1, 2$. *It is easy to check that with the presence of personalized reserves, no bidder can significantly overbid and win all auctions (otherwise she will incur large payments and thus violate their ROAS constraints), and therefore each bidder will obtain non-zero value. This aligns with our intuition presented in Sections 3 that states personalized reserves benefit individual welfare.*

*In the left subgraph of Figure 2, we color the region of all pairs of uniform bid multipliers* $(\alpha_1, \alpha_2) \in [1, \infty)^2$ *that induce feasible bid profiles* $(\boldsymbol{b}_1, \boldsymbol{b}_2) \in \mathcal{F}$, *where the blue dotted region corresponds to bid profiles under which bidder 1 wins only* $\mathcal{A}_1$, *and the grey vertically-dashed region corresponds to bid profiles under which bidder 1 wins* $\mathcal{A}_1$ *and* $\mathcal{A}_2$. *From this subgraph, we can see that* $\mathcal{F}^u_{-1}(\alpha_1 \boldsymbol{v}_1) = \{\alpha_2 \boldsymbol{v}_2 : \alpha_2 \in$ *any colored vertical line segments at* $\alpha_1\}$ *and similarly* $\mathcal{F}^u_{-2}(\alpha_2 \boldsymbol{v}_2) = \{\alpha_1 \boldsymbol{v}_1 : \alpha_1 \in$ *any colored horizontal line segments at* $\alpha_2\}$. *On the right subgraph of Figure 2, for each bidder* $i = 1, 2$, *we plot the individual welfare guarantee* $1 - \frac{1-\beta}{\alpha_i} \cdot \frac{\text{OPT}_{-i}}{\text{OPT}_i}$ *as well as* $\min_{\boldsymbol{b}_{-i} \in \mathcal{F}^u_{-i}(\alpha_i \boldsymbol{v}_i)} \frac{W_i(\mathcal{X}(\boldsymbol{b}))}{\text{OPT}_i}$ *which is the worst case welfare among all outcomes induced by uniform bid profiles that satisfy both bidders' ROAS constraints.*

On the left subgraph of Figure 2, we observe that it is easier for bidder 2 to ensure a non-empty feasibility set $\mathcal{F}^u_{-2}(\alpha_2 \boldsymbol{v}_2)$ at large $\alpha_2$ values than bidder 1 to ensure non-empty $\mathcal{F}^u_{-1}(\alpha_1 \boldsymbol{v}_1)$ at large $\alpha_1$; e.g. for large $\alpha_2$ such as $\alpha_2 = 3$, bidder 1 can take any $\alpha_1 \in [1, 1.75]$, but for large $\alpha_1 = 3$, bidder 2 can only take $\alpha_2 \in [1, 1.25]$. Nevertheless on the right subgraph, we see that bidder 2's realized welfare is much closer to her theoretical lower bound guarantee than that of bidder 1. Therefore this highlights a tradeoff between uniform multiplier feasibility and welfare guarantee.

## C.3   Proof for Theorem 4.2

THEOREM C.3 (RESTATEMENT OF THEOREM 4.2). *Consider 2 bidders competing in three SPA auctions whose values are indicated in the following table for any* $\beta \in (0, 1)$ *and* $y \geq 0$.

|          | Auction 1 | Auction 2 | Auction 3 . |
|----------|-----------|-----------|-------------|
| bidder 1 | $y$       | $v$       | $0$         |
| bidder 2 | $0$       | $v - \epsilon$ | $\gamma + \frac{1}{1-\beta} \cdot \epsilon$ |

*Bidder 1's multiplier is fixed to be $\alpha_1 > 1$, and consider $v = \frac{1-\beta}{\alpha_1 - 1} \cdot \gamma$ for any $\gamma > 0$. The reserve prices are set to be $r_{i,j} = \beta v_{i,j}$. Then, we have*

$$\min_{\boldsymbol{b} \in \mathcal{F}} \frac{W_1(\mathcal{X}(\boldsymbol{b}))}{OPT_1} = 1 - \frac{1 - \beta}{\alpha_1 - 1} \cdot \frac{OPT_{-1} - \frac{1}{1-\beta} \cdot \epsilon}{OPT_1} \tag{15}$$

*Taking $\epsilon \to 0$ shows that bidder 1's welfare is equal to the individual welfare guarantee in Theorem 4.1.*

**Remark C.2.** *We remark that as $\epsilon \to 0$, $\frac{OPT_{-i}}{OPT_i} = \frac{\frac{\alpha_1 - 1}{1-\beta} v}{y + v} \in \left[0, \frac{\alpha_1 - 1}{1-\beta}\right]$, so by varying $y \in [0, \infty)$, the above example demonstrates our individual welfare lower bound in Theorem 4.1 is tight for any nontrivial market share ratio $\frac{OPT_i}{OPT_{-i}} \in \left[\frac{\alpha_1 - 1}{1-\beta}, \infty\right)$.*

*proof* Note that in any feasible outcome, bidder 1 must win auction 1, and bidder 2 must win auction 3. Hence for auction 2, we only need to consider the following outcome:

**Bidder 1 loses auction 2, and suffers welfare loss** $v$. This outcome can be achieved by setting $\alpha_2$ such that $\alpha_2(v - \epsilon) > \alpha_1 v$. Bidder 2 accumulates value $v + \gamma + \left(\frac{1}{1-\beta} - 1\right)\epsilon$. Her payment for auction 2 is $\max\{\alpha_1 v, \beta(v - \epsilon)\}$, and for auction 3 is $\beta\left(\gamma + \frac{1}{1-\beta} \cdot \epsilon\right)$. The following shows that her ROAS constraint is satisfied:

$$v + \gamma + \left(\frac{1}{1-\beta} - 1\right)\epsilon - \max\{\alpha_1 v, \beta(v - \epsilon)\} - \beta\left(\gamma + \frac{1}{1-\beta} \cdot \epsilon\right)$$

$$\overset{(a)}{=} v + \gamma + \left(\frac{1}{1-\beta} - 1\right)\epsilon - \alpha_1 v - \beta\left(\gamma + \frac{1}{1-\beta} \cdot \epsilon\right)$$

$$= (1 - \alpha_1)v + (1 - \beta)\gamma + \left(\frac{1}{1-\beta} - 1 - \beta \cdot \frac{1}{1-\beta}\right)\epsilon$$

$$= 0,$$

where in (a) we used the fact $\beta \leq 1 < \alpha_1$ and $\epsilon \to 0$. In the final equality we used the definition that $v = \frac{1-\beta}{\alpha_1 - 1} \cdot \gamma$. On the other hand, bidder 1's ROAS constraint is apparently satisfied.

Under this outcome, denoting bidder 1's welfare as $W_1$ we have

$$\frac{W_1}{OPT_1} = 1 - \frac{v}{OPT_1} = 1 - \frac{1-\beta}{\alpha_i - 1} \cdot \frac{\gamma}{OPT_i} = 1 - \frac{1-\beta}{\alpha_i - 1} \cdot \frac{OPT_{-i} - \frac{1}{\beta} \cdot \epsilon}{OPT_i}$$

## C.4 Proof of Proposition C.1

PROOF. For simplicity, denote $\delta_{i,j} = OPT_{i,j} - W_{i,j}(\boldsymbol{x})$. Then, $OPT_i - W_i(\boldsymbol{x}) = \sum_{j \in [M]: \delta_{i,j} > 0} \delta_{i,j} + \sum_{j \in [M]: \delta_{i,j} = 0} \delta_{i,j} + \sum_{j \in [M]: \delta_{i,j} < 0} \delta_{i,j} = \text{LOSS}_i(\boldsymbol{x}) + \sum_{j \in [M]: W_{i,j}(\boldsymbol{x}) > OPT_{i,j}} \left(OPT_{i,j} - W_{i,j}(\boldsymbol{x})\right) \leq \text{LOSS}_i(\boldsymbol{x}) \leq B$. Rearranging and dividing both sides by $OPT_i$ we get $\frac{W_i(\boldsymbol{x})}{OPT_i} \geq 1 - \frac{B}{OPT_i}$.

Here we remark that it is possible to have $W_{i,j}(\boldsymbol{x}) > OPT_{i,j}$ because bidders may overbid, and therefore win auctions/slots that they would not have won under the efficient outcome. □

## C.5 Proof of Theorem 4.3

PROOF. Let $\boldsymbol{b} \in \mathcal{F}$ be any feasible bid profile, and let $\boldsymbol{x} = \mathcal{X}(\boldsymbol{b})$ be the corresponding outcome. Also, let $C_i(\boldsymbol{b})$ be the set of coverings defined in Eq. (8), and consider any $C \in C_i(\boldsymbol{x})$. Note that any competitor $k \in C$ must have $\mathcal{B}_i(k) \neq \emptyset$.

In Eq. (10) within the proof of Theorem 4.1, we showed $\text{LOSS}_i(\boldsymbol{x}) \leq \frac{1-\beta}{\alpha_i - 1} \sum_{k \in C} W_k(\boldsymbol{x})$, so

$$\text{LOSS}_i(\boldsymbol{x}) \leq \frac{1-\beta}{\alpha_i - 1} \sum_{k \in C} W_k(\boldsymbol{x}) \overset{(a)}{\leq} \frac{1-\beta}{\alpha_i - 1} \sum_{k \in [N]/\{i\}: \mathcal{B}_i(k) \neq \emptyset} W_k(\boldsymbol{x})$$

$$\leq \frac{1-\beta}{\alpha_i - 1} \sum_{k \in [N]/\{i\}: \mathcal{B}_i(k) \neq \emptyset} OPT_k,$$

Rearranging and applying Proposition C.1 yields the desired lower bound. □

# D ADDITIONAL MATERIALS FOR SECTION 5

## D.1 Additional Definitions and Lemmas for Section 5

The following lemma shows that for anonymous and truthful auctions, the probability of the lowest bidder winning a single auction is capped by a bound that decreases as the number of bidders grow.

**Lemma D.1** (Lemma 3 in [37]). *In an anonymous and truthful auction for a single item with $N$ bidders, the bidder who submits the lowest bid wins the item with probability at most $\frac{1}{N}$.*

The following technical definition and lemma (i.e. Definition D.1 and Lemma D.2) concerns the scenario where only one bidder participates in the auction (others bid 0), and present an upper bound on the probability and cost respectively for the single bidder to win the auction.

**Definition D.1** (Single bidder purchase probability and bid threshold). *For any allocation-anonymous and truthful auction $\mathcal{A}$, consider the setting with a single bidder who submits bid $b > 0$ and define*

$$\pi_{\mathcal{A}} = \lim_{b \to \infty} \mathbb{P}(\text{bidder wins item with bid } b), \tag{16}$$

*where the limit exists because in a truthful auction, $\mathbb{P}(\text{bidder wins item with bid } b)$ increases in $b$ (see Definition 2.1 for truthful auctions). Assume this max probability is reached at some bid threshold $Q_{\mathcal{A}}$, i.e.*

$$Q_{\mathcal{A}} = \min\{b > 0 : \mathbb{P}(\text{bidder wins item with bid } b) = \pi_{\mathcal{A}}\}. \tag{17}$$

Note that in a deterministic single-slot auction that allocates to the highest bidder, $\pi_{\mathcal{A}} = 1$, and $Q_{\mathcal{A}} \to 0$. For example, in an SPA with no reserve, the single bidder can win the auction with any arbitrarily small positive bid with probability 1.

**Lemma D.2** (Lemma 4 in [37]). *For any allocation-anonymous and truthful auction $\mathcal{A}$ with single-bidder purchase probability $\pi_{\mathcal{A}}$ and bid threshold $Q_{\mathcal{A}}$, the expected cost for a single bidder for winning the item is at most $\pi_{\mathcal{A}} \cdot Q_{\mathcal{A}}$.*

## D.2 Proof of Theorem 5.1

THEOREM D.3 (RESTATEMENT OF THEOREM 5.1). *For any auction $\mathcal{A}$ that is allocation-anonymous, truthful, and possibly randomized,* [5] *consider an autobidding problem instance w.r.t. $\mathcal{A}$ with $M = 2K + 1$ auctions and $N = K + 1$ bidders. Fix bidder 0's bid multiplier to be $\alpha_0$ and some $\beta \in [0, 1)$. Consider the bidder values $\{v_{i,j}\}_{i \in [N], j \in [M]}$ given in the following table.*

|       | $A_1$ | $A_2$ | ... | $A_K$ | $A_{K+1}$ | $A_{K+2}$ | ... | $A_{2K}$ | $A_{2K+1}$ |
|-------|-------|-------|-----|-------|-----------|-----------|-----|----------|------------|
| $B_1$ | $\frac{\alpha_0 v + \epsilon}{\rho}$ | $\frac{\alpha_0 v + 2\epsilon}{\rho}$ | ... | $\frac{\alpha_0 v + K\epsilon}{\rho}$ | $\gamma$ | $0$ | ... | $0$ | $0$ |
| $B_2$ | $\frac{\alpha_0 v + 2\epsilon}{\rho}$ | $\frac{\alpha_0 v + 3\epsilon}{\rho}$ | ... | $\frac{\alpha_0 v + \epsilon}{\rho}$ | $0$ | $\gamma$ | ... | $0$ | $0$ |
| $\vdots$ | $\vdots$ | $\vdots$ | | $\vdots$ | $\vdots$ | $\vdots$ | | $\vdots$ | $\vdots$ |
| $B_K$ | $\frac{\alpha_0 v + K\epsilon}{\rho}$ | $\frac{\alpha_0 v + \epsilon}{\rho}$ | ... | $\frac{\alpha_0 v + (K-1)\epsilon}{\rho}$ | $0$ | $0$ | ... | $\gamma$ | $0$ |
| $B_0$ | $v$ | $v$ | ... | $v$ | $0$ | $0$ | ... | $0$ | $y$ |

*In the table, we let $\gamma > \frac{Q_{\mathcal{A}}}{\beta} > Q_{\mathcal{A}}$, $\epsilon = O(1/K^3)$ and $v = \frac{1-\beta}{\alpha_0 - 1} \cdot \pi_{\mathcal{A}} \cdot \gamma$. Let $\rho, y$ and a large enough $K$ satisfy the following:*

$$\alpha_0 < \rho < \frac{\alpha_0}{\beta} \ s.t. \ \frac{\alpha_0 v + K\epsilon}{\rho} < v, \quad \text{and} \quad y > \max\left\{\frac{Q_{\mathcal{A}}}{\alpha_0}, \frac{\alpha_0 v}{\pi_{\mathcal{A}}}\right\}, \tag{18}$$

*where $Q_{\mathcal{A}}, \pi_{\mathcal{A}}$ are defined in Definition D.1. Further, suppose the platform enforces personalized reference prices $\mathbf{r} \in \mathbb{R}_+^{N \times M}$ on top of auction $\mathcal{A}$, where $r_{i,j} = \beta v_{i,j}$. Then, letting the (possibly random) outcome be $\mathbf{x}$ when bidders $1, \dots K$ all adopt the bid multiplier $\rho$, the ROAS constraints for all bidders are satisfied when $K \to \infty$ and $\rho \to \alpha_0$, and for bidder 0 we have*

$$\lim_{K \to \infty} \frac{\mathbb{E}_{\mathcal{A}}[W_0(\mathbf{x})]}{\mathbb{E}_{\mathcal{A}}[OPT_0]} \le 1 - \frac{1-\beta}{\alpha_0 - 1} \cdot \lim_{K \to \infty} \frac{\mathbb{E}_{\mathcal{A}}[OPT_{-0}]}{\mathbb{E}_{\mathcal{A}}[OPT_0]} \tag{19}$$

*where $\mathbb{E}_{\mathcal{A}}$ is taken w.r.t. the randomness in outcome $\mathbf{x}$ due to randomness in the auction $\mathcal{A}$.*

---

[5]Here, we assume all auctions of interest are individually rational (IR), i.e. the payment of a bidder is always less than her submitted bid.

PROOF. First note that bidder 0 only has competition in auctions $A_1 \ldots A_K$, and hence can only incur a loss (that contributes to $\text{LOSS}_0(\boldsymbol{x})$ defined in Equations (6)) within these auctions. Hence $\mathbb{E}_{\mathcal{A}}[\text{LOSS}_0(\boldsymbol{x})] = v \sum_{j \in [K]} \mathbb{P}(\text{bidder 0 loses auction } j)$. Then we consider the following:

$$
\begin{aligned}
\mathbb{E}_{\mathcal{A}}[\text{LOSS}_0(\boldsymbol{x})] &= v \sum_{j \in [K]} \mathbb{P}(\text{bidder 0 loses auction } j) = v \sum_{j \in [K]} (1 - \mathbb{P}(\text{bidder 0 wins auction } j)) \\
&\overset{(a)}{\geq} v \cdot \frac{K^2}{K+1} = \frac{1-\beta}{\alpha_0 - 1} \cdot \gamma \cdot \pi_{\mathcal{A}} \cdot \frac{K^2}{K+1} \overset{(b)}{=} \frac{1-\beta}{\alpha_0 - 1} \cdot \mathbb{E}_{\mathcal{A}}[\text{OPT}_{-0}] \cdot \frac{K}{K+1} .
\end{aligned}
\tag{20}
$$

Here (a) holds because bidder 0 bids $\alpha_0 v$ for any auction in $1,2 \ldots K$, which is strictly less than all other bidders' bids as they all adopt multipliers $\rho$ in these auctions, so from Lemma D.1, we have $\mathbb{P}(\text{bidder 0 wins auction } j) \leq \frac{1}{K+1}$; in (b) we used the fact that $\mathbb{E}_{\mathcal{A}}[\text{OPT}_{-0}] = \sum_{j=K+1}^{2K} \mathbb{E}_{\mathcal{A}}[\gamma] = \gamma \cdot K \cdot \pi_{\mathcal{A}}$ since there is only a single non-zero bidder in auctions $A_{K+1} \ldots A_{2K}$ and each bidder submits a bid $\rho\gamma > \rho > Q_{\mathcal{A}}$ (see Definition D.1).

Therefore we have

$$
\lim_{K \to \infty} \frac{\mathbb{E}_{\mathcal{A}}[W_0(\boldsymbol{x})]}{\mathbb{E}_{\mathcal{A}}[\text{OPT}_0]} \overset{(a)}{=} 1 - \lim_{K \to \infty} \frac{\mathbb{E}_{\mathcal{A}}[\text{LOSS}_0(\boldsymbol{x})]}{\mathbb{E}_{\mathcal{A}}[\text{OPT}_0]} \leq 1 - \frac{1-\beta}{\alpha_0 - 1} \lim_{K \to \infty} \frac{\mathbb{E}_{\mathcal{A}}[\text{OPT}_{-0}]}{\mathbb{E}_{\mathcal{A}}[\text{OPT}_0]} ,
\tag{21}
$$

where (a) follows from the fact that in our constructed autobidding instance, bidder 0's acquired value in each auction cannot exceed that under the efficient allocation, and hence can only incur loss in welfare.

Now it only remains to show that the multiplies $(\alpha_0, \rho, \ldots \rho) \in (1, \infty)^{K+1}$ yields a feasible outcome, i.e. the ROI constraints of each bidder is satisfied in expectation. Let $V_{i,j}$ and $C_{i,j}$ be the expected value and cost of bidder $i$ in auction $A_j$, respectively.

**1. Showing bidder 0's ROI constraint is satisfied**. We show by the following: bidder 0 only incurs a non-zero expected cost in auctions $A_1 \ldots A_K$ and $A_{2K+1}$, and we will show that the expected value $V_{0,2K+1}$ is lower bounded by the expected costs $C_{0,2K+1} + \sum_{j \in [K]} C_{0,j}$.

Since $\alpha_0 y > Q_{\mathcal{A}}$, the definition of the single-bidder purchasing probability in Definition D.1 implies that bidder 0 acquires an expected value from auction $A_{2K+1}$ of $V_{0,2K+1} = \pi_{\mathcal{A}} y$. Further, since bidder 0 is submits the lowest bids in auctions $A_1 \ldots A_K$ under bid multiplier profile $(\alpha_0, \rho \ldots \rho) \in (0, \infty)^{K+1}$, from Lemma D.1, we have $\mathbb{P}(\text{bidder 0 wins auction } j) \leq \frac{1}{K+1}$ for all $j \in [K]$. Since the payment of a bidder in an auction is at most her submitted bid (as the auction is IR), we know that $\sum_{j \in [K]} C_{0,j} \leq K \cdot \frac{\alpha_0 v}{K+1} < \pi_{\mathcal{A}} y = V_{0,2K+1}$, where the inequality follows from the definition of $y$ in Equation (18) such that $y > \max\left\{\frac{Q_{\mathcal{A}}}{\alpha_0}, \frac{\alpha_0 v}{\pi_{\mathcal{A}}}\right\}$. This implies bidder 0's ROI constraint is satisfied.

**2. Showing bidder $i$'s ROI constraint is satisfied for any** $i = 1, 2 \ldots K$. We show this by considering the following: bidder $i$ only incurs a non-zero expected cost in auctions $A_1 \ldots A_K$ and $A_{K+i}$, and we will show that the expected values $V_{i,K+i} + \sum_{k \in [K]} V_{i,j}$ is lower bounded by the expected costs $C_{i,K+i} + \sum_{j \in [K]} C_{i,j}$.

- **Calculate cost $C_{i,K+i}$:** For auction $A_{K+i}$, bidder $i$'s bid is $\rho\gamma > \gamma > Q_{\mathcal{A}}$ from the definition of $\gamma$, so by Definition D.1, the probability of $i$ winning the item in auction $A_{K+i}$ is $\pi_{\mathcal{A}}$, and the expected cost is

$$
C_{i,K+i} \leq \pi_{\mathcal{A}} \cdot \max\{r_{i,K+i}, Q_{\mathcal{A}}\} \leq \pi_{\mathcal{A}} \cdot \beta\gamma ,
\tag{22}
$$

  where the final inequality follows from the definition $r_{i,K+i} = \beta\gamma$
- **Upper bound costs $\sum_{j \in [K]} C_{i,j}$:** For auctions $[K] = 1 \ldots K$, bidder $i$'s total expected cost can be bounded as

$$
\begin{aligned}
\sum_{j \in [K]} C_{i,j} &\leq \rho \sum_{j \in [K]} v_{i,j} \mathbb{P}(\text{bidder } i \text{ wins auction } A_j) \\
&= \alpha_0 v \sum_{j \in [K]} \mathbb{P}(\text{bidder } i \text{ wins auction } A_j) + \frac{(K+1)K}{2}\epsilon .
\end{aligned}
\tag{23}
$$

  where the first inequality follows from a bidder's payment is at most her submitted bid since the auction is IR.
- **Calculate $V_{i,K+i}$:** Considering auction $A_{K+i}$, bidder $i$ is the only bidder, and since $\rho\gamma > \gamma > Q_{\mathcal{A}}$, the definition of the single-bidder purchasing probability in Definition D.1 implies that bidder $i$'s acquires an expected value from this auction of

$$
V_{i,K+i} = \pi_{\mathcal{A}} \cdot \gamma .
\tag{24}
$$

- **Lower bound $\sum_{k \in [K]} V_{i,j}$:**

$$
\sum_{k \in [K]} V_{i,j} \geq \frac{\alpha_0 v}{\rho} \sum_{j \in [K]} \mathbb{P}(\text{bidder } i \text{ wins auction } j) .
\tag{25}
$$

Combining Equations (22),(23),(24) and (25), we get

$$
\sum_{j \in [K]} V_{i,j} + V_{i,K+i} - \left( \sum_{j \in [K]} C_{i,j} + C_{i,K+i} \right)
$$

$$
\geq \pi_{\mathcal{A}} \cdot \gamma + \frac{\alpha_0 v}{\rho} \cdot \sum_{j \in [K]} \mathbb{P} \left( \text{bidder } i \text{ wins auction } j \right)
$$

$$
- \left( \pi_{\mathcal{A}} \cdot \beta \gamma + \alpha_0 v \cdot \sum_{j \in [K]} \mathbb{P} \left( \text{bidder } i \text{ wins auction } j \right) + \frac{(K+1)K}{2} \epsilon \right)
$$

$$
= \pi_{\mathcal{A}} \cdot (1 - \beta) \gamma - \left( \alpha_0 - \frac{\alpha_0}{\rho} \right) v \cdot \sum_{j \in [K]} \mathbb{P} \left( \text{bidder } i \text{ wins auction } j \right) - \frac{(K+1)K}{2} \epsilon \tag{26}
$$

$$
\overset{(a)}{=} (\alpha_0 - 1) v - \left( \alpha_0 - \frac{\alpha_0}{\rho} \right) v \cdot \sum_{j \in [K]} \mathbb{P} \left( \text{bidder } i \text{ wins auction } j \right) - \frac{(K+1)K}{2} \epsilon
$$

$$
\overset{(b)}{\geq} (\alpha_0 - 1) v - \left( \alpha_0 - \frac{\alpha_0}{\rho} \right) v - \frac{(K+1)K}{2} \epsilon
$$

,

where (a) follows from the definition $v = \frac{1-\beta}{\alpha_0 - 1} \cdot \pi_{\mathcal{A}} \cdot \gamma$; In (b) we used the fact that $\rho > \alpha_0 > 1$ and $\sum_{j \in [K]} \mathbb{P} \left( \text{bidder } i \text{ wins auction } A_j \right) \leq 1$ due to the following: Consider the set of bid values $\mathcal{B} = \{\alpha_0 v, \alpha_0 v + \epsilon, \alpha_0 v + 2\epsilon \ldots \alpha_0 v + K\epsilon\} \subseteq \mathbb{R}_{>0}$, and we recognize that any bid value $b_k \in \mathcal{B}$ exceeds the maximim reserve price $\beta v$ in auctions $A_1 \ldots A_K$. Therefore the constructed reserve prices do not affect allocation, and hence by anonymity of auction $\mathcal{A}$ there exists probabilities $\boldsymbol{q}(\mathcal{B}) = (q_0(\mathcal{B}), q_1(\mathcal{B}) \ldots q_K(\mathcal{B})) \in [0,1]^{K+1}$ where

$$
q_k(\mathcal{B}) = \mathbb{P}(\text{bid value } b_k \text{ wins auction } \mathcal{A} \text{ given competing bids } \boldsymbol{b}_{-k}) \quad \text{and} \quad \sum_{k=0}^{K} q_k(\mathcal{B}) \leq 1.
$$

We recognize that in each auction $A_1 \ldots A_K$, under bid multipliers $(\alpha_0, \rho, \ldots \rho) \in (1, \infty)^{K+1}$ the submitted bid profile is a cyclic permutation of $\mathcal{B}$. Therefore we know that

$$
\sum_{j \in [K]} \mathbb{P} \left( \text{bidder } i \text{ wins auction } j \right) = \sum_{k=1}^{K} q_k(\mathcal{B}) \leq 1 - q_0(\mathcal{B}) \leq 1
$$

Finally, by taking $\rho \to \alpha_0$ and $K \to \infty$ in Equation (26), and utilizing $\epsilon = O(1/K^3)$ we have

$$
\lim_{\rho \to \alpha_0} \lim_{K \to \infty} \sum_{j \in [K]} V_{i,j} + V_{i,K+i} - \left( \sum_{j \in [K]} C_{i,j} + C_{i,K+i} \right) \geq 0 \,.
$$

This shows that bidder $i$'s ROI constraint is satisfied. □

## E PROOFS FOR SECTION 6

### E.1 Proof of Theorem 6.2

PROOF. For convenience, define $\delta = 2 - \frac{1}{\Delta}$, so $\Delta > 1$ implies $\delta \in (1, 2)$, and further $1 > \beta > \frac{\Delta}{2\Delta - 1}$ implies $\frac{1}{\delta} < \beta < 1$.

Fix a bidder $i \in [K]$ and any feasible competing bid profile $\boldsymbol{b} \in \mathcal{U}$. Denote the corresponding outcome as $\boldsymbol{x} = \mathcal{X}(\boldsymbol{b})$, where $\boldsymbol{x} = (\boldsymbol{x}_1 \ldots \boldsymbol{x}_M)$ where $\boldsymbol{x}_j \in \{0, 1\}^{N \times L_j}$ is the outcome vector in auction $\mathcal{A}_j$. Note that by definition of $\mathcal{U}$ which is the set of undominated and feasible bids, under the outcome $\boldsymbol{x}$ all bidders' ROAS constraints are satisfied. Denote $\ell_{k,j}, \ell_{k,j}^*$ to be the position of bidder $k \in [N]$ in auction $j \in [M]$ under outcome $\boldsymbol{x}$ and the efficient outcome, respectively.

Recall in Eq.(8) the definition for the set of all "coverings" for bidder $i$, denoted as $C_i(\boldsymbol{x})$:

$$
\mathcal{B}_i(k; \boldsymbol{x}) = \left\{ j \in [M] : \text{OPT}_{i,j} > 0, \ v_{k,j} < v_{i,j} \text{ and } \ell_{k,j} \leq \ell_{i,j}^* < \ell_{i,j} \right\}
$$

$$
C_i(\boldsymbol{x}) = \left\{ C \subseteq [N]/\{i\} : (\mathcal{B}_i(k; \boldsymbol{x}))_{k \in C} \text{ is a maximal set cover of } \mathcal{L}_i(\boldsymbol{x}) \right\}
$$

where $\mathcal{L}_i(\boldsymbol{x}) = \{j \in [M] : W_{i,j}(\boldsymbol{x}) < \text{OPT}_{i,j}\}$ is the set of auctions in which bidder $i$'s acquired welfare is less than that of her welfare under the efficient outcome; see Definition C.1.

Denote $p_{k,j}$ as the payment of any bidder $k$, and $\hat{b}_{\ell,j}$ as the $\ell$th largest bid in any auction $j \in [M]$. Similar to the proof of Theorem 4.1, fix any covering $C \subseteq C_i(\boldsymbol{x})$, and any bidder $k \in C$, such that in some auction $j \in \mathcal{B}_i(k; \boldsymbol{x})$, we have $v_{k,j} < v_{i,j}$ but $\ell_{k,j} \leq \ell_{i,j}^* < \ell_{i,j}$. Thus

following a similar deduction as Eq. (12) in the proof of Theorem 4.1, bidder $k$'s payment is lower bounded as

$$
\text{For } j \in \mathcal{B}_i(k; \boldsymbol{x}), \quad p_{k,j} \overset{(a)}{\geq} \sum_{\ell=\ell_{k,j}}^{L_j} \left( \mu(\ell) - \mu(\ell+1) \right) \hat{b}_{\ell+1,j}
$$

$$
= \sum_{\ell=\ell_{k,j}}^{\ell_{i,j}-1} \left( \mu(\ell) - \mu(\ell+1) \right) \hat{b}_{\ell+1,j} + p_{i,j}
$$

$$
\overset{(b)}{\geq} \beta \left( \mu(\ell_{k,j}) - \mu(\ell_{i,j}) \right) v_{i,j} + \beta \cdot \mu(\ell_{i,j}) v_{i,j}
$$

$$
= \beta \mu(\ell_{k,j}) \cdot v_{i,j}
$$

$$
= \mu(\ell_{k,j}) v_{i,j} + \left( \beta - \frac{1}{\delta} \right) \left( \mu(\ell_{i,j}^*) - \mu(\ell_{i,j}) \right) v_{i,j} - (1-\beta) \cdot \mu(\ell_{i,j}) v_{i,j}
$$
$$
\quad - (1-\beta) \mu(\ell_{k,j}) v_{i,j} + \left( \frac{1}{\delta} - \beta \right) \mu(\ell_{i,j}^*) v_{i,j} + \left( 1 - \frac{1}{\delta} \right) \mu(\ell_{i,j}) v_{i,j} \tag{27}
$$

$$
\overset{(c)}{\geq} \mu(\ell_{k,j}) v_{i,j} + \left( \beta - \frac{1}{\delta} \right) \left( \mu(\ell_{i,j}^*) - \mu(\ell_{i,j}) \right) v_{i,j} - (1-\beta) \cdot \mu(\ell_{i,j}) v_{i,j}
$$
$$
\quad - \left( 1 - \frac{1}{\delta} \right) \mu(\ell_{k,j}) v_{i,j} + \left( 1 - \frac{1}{\delta} \right) \mu(\ell_{i,j}) v_{i,j}
$$

$$
= \left( \beta - \frac{1}{\delta} \right) \left( \mu(\ell_{i,j}^*) - \mu(\ell_{i,j}) \right) v_{i,j} - (1-\beta) \cdot \mu(\ell_{i,j}) v_{i,j}
$$
$$
\quad + \frac{1}{\delta} \mu(\ell_{k,j}) v_{i,j} + \left( 1 - \frac{1}{\delta} \right) \mu(\ell_{i,j}) v_{i,j}
$$

Here , (a) follows from the fact that for a fix bid profile, the payment of GSP or GFP for each bidder in an auction dominates that of VCG (see Example B.2 and discussions thereof); (b) follows from $\hat{b}_{\ell,j} \geq b_{i,j}$ for $\ell \leq \ell_{i,j}$, and since $\boldsymbol{b} \in \mathcal{U} \subseteq \mathbb{R}_+^{N \times M}$ is an undominated bid profile, Lemma 6.1 applies and $b_{i,j} \geq \beta v_{i,j}$. Also $p_{i,j} \geq r_{i,j} \geq \beta v_{i,j}$ be the definition of $\beta$-approximate reserves; (c) follows from the fact that $\beta > \frac{1}{\delta}$ and $\mu(\ell_{i,j}^*) \leq \mu(\ell_{k,j})$ since $\ell_{k,j} \leq \ell_{i,j}^*$ for any $k \in C \subseteq C_i(\boldsymbol{x})$ and $j \in \mathcal{B}_i(k; \boldsymbol{x})$; see definition in Eq. (8).

On the other hand, we have

$$
\sum_{j \in \mathcal{B}_i(k; \boldsymbol{x})} p_{k,j} + \sum_{j \notin \mathcal{B}_i(k; \boldsymbol{x})} p_{k,j} \leq \sum_{j \in \mathcal{B}_i(k; \boldsymbol{x})} \mu(\ell_{k,j}) v_{k,j} + \sum_{j \notin \mathcal{B}_i(k; \boldsymbol{x})} \mu(\ell_{k,j}) v_{k,j}
$$
$$
p_{k,j} \geq \beta \cdot \mu(\ell_{k,j}) v_{k,j} \quad \forall j \in [M],
$$

where the first inequality follows from bidder $k$'s ROAS constraint; the second inequality follows from the fact that any winning bidder's payment must be greater than her $\beta$-approximate reserves.

Combining the above inequalities and rearranging we get

$$
\sum_{j \in \mathcal{B}_i(k; \boldsymbol{x})} p_{k,j} \leq \sum_{j \in \mathcal{B}_i(k; \boldsymbol{x})} \mu(\ell_{k,j}) v_{k,j} + (1-\beta) \cdot \sum_{j \notin \mathcal{B}_i(k; \boldsymbol{x})} \mu(\ell_{k,j}) v_{k,j}, \tag{28}
$$

Summing Eq.(27) over all $j \in \mathcal{B}_i(k; \boldsymbol{x})$ and combining with Eq. (28), we get

$$
\left( \beta - \frac{1}{\delta} \right) \cdot \sum_{j \in \mathcal{B}_i(k; \boldsymbol{x})} \left( \mu(\ell_{i,j}^*) - \mu(\ell_{i,j}) \right) v_{i,j}
$$
$$
\leq (1-\beta) \cdot \left( \sum_{j \in \mathcal{B}_i(k; \boldsymbol{x})} \mu(\ell_{i,j}) v_{i,j} + \sum_{j \notin \mathcal{B}_i(k; \boldsymbol{x})} \mu(\ell_{k,j}) v_{k,j} \right) \tag{29}
$$
$$
+ \underbrace{\sum_{j \in \mathcal{B}_i(k; \boldsymbol{x})} \mu(\ell_{k,j}) v_{k,j} - \frac{1}{\delta} \sum_{j \in \mathcal{B}_i(k; \boldsymbol{x})} \mu(\ell_{k,j}) v_{i,j} - \left( 1 - \frac{1}{\delta} \right) \sum_{j \in \mathcal{B}_i(k; \boldsymbol{x})} \mu(\ell_{i,j}) v_{i,j}}_{Y}.
$$

We now upper bound $Y$:

$$\sum_{j \in \mathcal{B}_i(k;\boldsymbol{x})} \mu(\ell_{k,j}) v_{k,j} - \frac{1}{\delta} \sum_{j \in \mathcal{B}_i(k;\boldsymbol{x})} \mu(\ell_{k,j}) v_{i,j} - \left(1 - \frac{1}{\delta}\right) \sum_{j \in \mathcal{B}_i(k;\boldsymbol{x})} \mu(\ell_{i,j}) v_{i,j}$$

$$= \left(1 - \frac{1}{\delta}\right) \sum_{j \in \mathcal{B}_i(k;\boldsymbol{x})} \mu(\ell_{k,j}) v_{k,j} - \frac{1}{\delta} \sum_{j \in \mathcal{B}_i(k;\boldsymbol{x})} \mu(\ell_{k,j}) \left(v_{i,j} - v_{k,j}\right) - \left(1 - \frac{1}{\delta}\right) \sum_{j \in \mathcal{B}_i(k;\boldsymbol{x})} \mu(\ell_{i,j}) v_{i,j}$$

$$= \left(1 - \frac{1}{\delta}\right) \sum_{j \in \mathcal{B}_i(k;\boldsymbol{x})} \mu(\ell_{k,j}) \left(v_{k,j} - v_{i,j}\right) - \frac{1}{\delta} \sum_{j \in \mathcal{B}_i(k;\boldsymbol{x})} \mu(\ell_{k,j}) \left(v_{i,j} - v_{k,j}\right)$$

$$\quad + \left(1 - \frac{1}{\delta}\right) \sum_{j \in \mathcal{B}_i(k;\boldsymbol{x})} \left(\mu(\ell_{k,j}) - \mu(\ell_{i,j})\right) v_{i,j}$$

$$= \left(1 - \frac{1}{\delta}\right) \sum_{j \in \mathcal{B}_i(k;\boldsymbol{x})} \mu(\ell_{k,j}) \left(v_{k,j} - v_{i,j}\right) + \sum_{j \in \mathcal{B}_i(k;\boldsymbol{x})} \frac{\mu(\ell_{k,j}) v_{i,j}}{\delta} \left((\delta - 1)\left(1 - \frac{\mu(\ell_{i,j})}{\mu(\ell_{k,j})}\right) - \left(1 - \frac{v_{k,j}}{v_{i,j}}\right)\right) \qquad (30)$$

$$\overset{(a)}{\leq} \left(1 - \frac{1}{\delta}\right) \sum_{j \in \mathcal{B}_i(k;\boldsymbol{x})} \mu(\ell_{k,j}) \left(v_{k,j} - v_{i,j}\right) + \sum_{j \in \mathcal{B}_i(k;\boldsymbol{x})} \frac{\mu(\ell_{k,j}) v_{i,j}}{\delta} \left((\delta - 1) - \left(1 - \frac{v_{k,j}}{v_{i,j}}\right)\right)$$

$$= \left(1 - \frac{1}{\delta}\right) \sum_{j \in \mathcal{B}_i(k;\boldsymbol{x})} \mu(\ell_{k,j}) \left(v_{k,j} - v_{i,j}\right) + \sum_{j \in \mathcal{B}_i(k;\boldsymbol{x})} \frac{\mu(\ell_{k,j}) v_{i,j}}{\delta} \left((\delta - 2) + \frac{v_{k,j}}{v_{i,j}}\right)$$

$$\overset{(b)}{\leq} \left(1 - \frac{1}{\delta}\right) \sum_{j \in \mathcal{B}_i(k;\boldsymbol{x})} \mu(\ell_{k,j}) \left(v_{k,j} - v_{i,j}\right)$$

$$\overset{(c)}{\leq} (1 - \beta) \sum_{j \in \mathcal{B}_i(k;\boldsymbol{x})} \mu(\ell_{k,j}) \left(v_{k,j} - v_{i,j}\right).$$

where in (a) we recall $\delta > 1$ and $\ell_{k,j} < \ell_{i,j}$ for any $k \in C$ and $j \in \mathcal{B}_i(k;\boldsymbol{x})$ so that $\mu(\ell_{k,j}) > \mu(\ell_{i,j})$; (b) follows from the fact that values are $\delta$-separated, so $v_{i,j} > v_{k,j}$ for $k \in C$ and $j \in \mathcal{B}_i(k;\boldsymbol{x})$ implies $\frac{v_{k,j}}{v_{i,j}} \leq \frac{1}{\Delta} = 2 - \delta$; in (c) we used the fact that $\beta > \frac{1}{\delta}$ so $1 - \beta < 1 - \frac{1}{\delta}$, and the fact that $v_{k,j} < v_{i,j}$ for any $k \in C$ and $j \in \mathcal{B}_i(k;\boldsymbol{x})$.

Combining Equations (29) and (30) we get

$$\left(\beta - \frac{1}{\delta}\right) \cdot \sum_{j \in \mathcal{B}_i(k;\boldsymbol{x})} \left(\mu(\ell_{i,j}^*) - \mu(\ell_{i,j})\right) v_{i,j}$$

$$\leq (1 - \beta) \cdot \left(\sum_{j \in \mathcal{B}_i(k;\boldsymbol{x})} \mu(\ell_{i,j}) v_{i,j} + \sum_{j \notin \mathcal{B}_i(k;\boldsymbol{x})} \mu(\ell_{k,j}) v_{k,j} + \sum_{j \in \mathcal{B}_i(k;\boldsymbol{x})} \mu(\ell_{k,j}) \left(v_{k,j} - v_{i,j}\right)\right)$$

$$= (1 - \beta) \cdot \left(\sum_{j \in \mathcal{B}_i(k;\boldsymbol{x})} \mu(\ell_{i,j}) v_{i,j} + \sum_{j \in [M]} \mu(\ell_{k,j}) v_{k,j} - \sum_{j \in \mathcal{B}_i(k;\boldsymbol{x})} \mu(\ell_{k,j}) v_{i,j}\right) \qquad (31)$$

$$\overset{(a)}{\leq} (1 - \beta) \cdot \left(\sum_{j \in \mathcal{B}_i(k;\boldsymbol{x})} \mu(\ell_{i,j}) v_{i,j} + \sum_{j \in [M]} \mu(\ell_{k,j}) v_{k,j} - \sum_{j \in \mathcal{B}_i(k;\boldsymbol{x})} \mu(\ell_{i,j}^*) v_{i,j}\right).$$

$$\implies \sum_{j \in \mathcal{B}_i(k;\boldsymbol{x})} \left(\mu(\ell_{i,j}^*) - \mu(\ell_{i,j})\right) v_{i,j} \leq \frac{1 - \beta}{1 - \frac{1}{\delta}} \sum_{j \in [M]} \mu(\ell_{k,j}) v_{k,j}$$

where (a) follows from $\mu(\ell_{i,j}^*) \leq \mu(\ell_{k,j})$ due to the fact that $\ell_{k,j} < \ell_{i,j}$ for any $k \in C$ and $j \in \mathcal{B}_i(k;\boldsymbol{x})$.

Summing the above over all $k \in C$, and following the same arguments as in Eq.(10) of the proof of Theorem 4.1, we have

$$\text{LOSS}_i(\boldsymbol{x}) = \sum_{j \in \mathcal{L}_i(\boldsymbol{x})} \left( \mu(\ell_{i,j}^*) - \mu(\ell_{i,j}) \right) v_{i,j}$$

$$\leq \sum_{k \in C} \sum_{j \in \mathcal{B}_i(k;\boldsymbol{x})} \left( \mu(\ell_{i,j}^*) - \mu(\ell_{i,j}) \right) v_{i,j}$$

$$\leq \frac{1-\beta}{1-\frac{1}{\delta}} \sum_{k \in C} \sum_{j \in [M]} \mu(\ell_{k,j}) v_{k,j}$$

$$= \frac{1-\beta}{1-\frac{1}{\delta}} \sum_{k \in C} W_k(\boldsymbol{x}) \tag{32}$$

$$\leq \frac{1-\beta}{1-\frac{1}{\delta}} W_{-i}(\boldsymbol{x})$$

$$\leq \frac{1-\beta}{1-\frac{1}{\delta}} \left( \text{OPT}_{-i} + \text{LOSS}_i(\boldsymbol{x}) \right).$$

Rearranging we get $\text{LOSS}_i(\boldsymbol{x}) \leq \frac{1-\beta}{\beta-\frac{1}{\delta}} \text{OPT}_{-i} = \frac{1-\beta}{\beta-\frac{\Delta}{2\Delta-1}} \text{OPT}_{-i}$. Finally, applying Proposition C.1 w.r.t. upper bound of $\text{LOSS}_i(\boldsymbol{x})$ and using the fact that the competing bid profile is arbitrary, we obtain the desired welfare guarantee lower bound. $\square$

# F  ADDITIONAL MATERIALS FOR SECTION 7

## F.1  Calculating uniform bid multipliers.

Here we describe the procedure to generate bid multipliers for bidders in two scenarios, one with reserve prices set with ML advice we generated earlier, and the other without reserve price which we call the "control experiment" (for consistency we let $\beta = 0$ correspond to this no-reserve price setup). We first describe the procedure to generate bid multipliers for the scenario with reserve prices. In particular, we calculate each advertiser's uniform bid multiplier using gradient descent to emulate uniform bidding practices in reality, because descent/primal-dual methods have been widely adopted for real world autobidding and has proven to have near-optimal convergence and performance guarantees (see e.g. [1, 4, 39]. Formally, for each accuracy level $\beta \in \{0.25, 0.5, 0.75\}$, we run $2T$ rounds of gradient descent, where in each round we keep the auction environment, including advertiser values, number of slots and CTRs the same as those we derived from the aforementioned semi-synthetic data.

The first $T$ rounds are dedicated to "warm start" our treatments for reserve prices with different accuracy level: we simulate VCG auctions without reserves until all bidders' uniform bid multipliers convergence. In particular, with an initial uniform bid multiplier $\alpha_{i,1}$ for bidder $i$, for each round $t \in [T]$, set advertiser $i$'s bid multiplier to be $\alpha_{i,t}$, and run $M$ VCG auctions without reserves, where values are $\boldsymbol{v} = (v_{i,j})_{i \in [N], j \in [M]}$, ad slot numbers are $(L_j)_{j \in [M]}$ and corresponding CTRs are $(\boldsymbol{\mu}_j)_{j \in [M]}$. For round $t \in [T]$, denoting $w_{i,t}$ and $p_{i,t}$ as the total realized welfare and payment for bidder $i$ across all $M$ auctions, we update the uniform bid multiplier with gradient descent in the log-space (note that a similar approach has been used in [3, 16])

$$\log \alpha_{i,t+1} = (1 - \eta_t) \log \alpha_{i,t} + \eta_t \log(w_{i,t}/p_{i,t}). \tag{33}$$

Here $\eta_t$ are properly chosen step-sizes that ensure convergence within $T$ rounds. Intuitively, when $w_{i,t} > p_{i,t}$, the per-round ROAS balance for bidder $i$ is positive, so in the next round the bidder would have leeway to bid more aggressively with larger bid multipliers to acquire more welfare. We take the bid-multiplier value $\alpha_{i,T+1}$ for $\beta \in \{0, 0.25, 0.5, 0.75\}$ to be our initial uniform bid multipliers for treatments of VCG auctions with personalized reserves. For the next $T$ rounds, we repeat the above procedure for running VCG auctions, but with personalized reserve prices $r_{i,j} = \underline{v}_{i,j}^\beta$ for treatment trials corresponding to $\beta \in \{0.25, 0.5, 0.75\}$, respectively, where we compute $(\alpha_{i,t}^\beta)_{i \in [N], t=T+1\ldots 2T}$ according to Eq. (33). Finally, we arrive at our bid multipliers $\alpha_i^\beta := \alpha_{i,2T}^\beta$.

We repeat the above procedure for our control experiment with no reserve price, and again arrive at bid multipliers $\alpha_i^\beta := \alpha_{i,2T}^\beta$ for $\beta = 0$.

