# OpenReview forum: "Individual Welfare Guarantees in the Autobidding World with Machine-learned Advice"
_ACM.org/TheWebConf/2024/Conference — TheWebConf24 Oral_

### Official Review · Reviewer_ArkB · 2023-11-15

**Novelty:** 3
**Technical Quality:** 3

**Review:**

Summary: The paper analyzes an individual bidder’s welfare loss in autobidding auctions with and without machine-learned advice, delving into how advertiser strategies are linked to such losses. It showcases how ad platforms can enhance welfare guarantees, both at an aggregate and individual bidder level, by integrating ML advice as personalized reserve prices for autobidders adhering to a return on ad spend (ROAS) constraint.

Major issues:

Overall, the paper appears ok to me, but the technical contributions to existing works aren't entirely clear to me. Despite the authors mentioning the need for "novel analyses on the value-expenditure trade-offs" (lines 185-187), I struggle to appreciate the novelty. The statement of Theorem 4.1 seems lengthy, and the final results rely heavily on the definitions and metrics introduced in Section 2. I'm uncertain about the impact of the new metrics, specifically the individual welfare metric in Section 2, and the role of ROAS constraints in the proofs. Clarifying how much these constraints can be relaxed and their specific role in the proofs would be beneficial. Although the authors mention in Footnote 3 that the results apply to broader ROI settings, I would appreciate more detailed discussions on this aspect.

Minor issues:

The presentation is satisfactory, but it's advisable to temper the emphasis on the paper's contributions. There's frequent use of the term "novel," and the phrase "the proof techniques require novel analyses" might sound overly self-assured. I am not sure why authors are so confidence that there is no trivial or simpler approach to complete the proofs there.

**Questions:**

Could you please add more elaborations on the questions raised in major issues?

**Reviewer Confidence:**

3: The reviewer is confident but not certain that the evaluation is correct

**Scope:**

4: The work is relevant to the Web and to the track, and is of broad interest to the community

---

### Official Review · Reviewer_pkrx · 2023-11-17

**Novelty:** 5
**Technical Quality:** 5

**Review:**

Pros:

- This paper considers an important question on the individual welfare guarantees with auto-bidders when the mechanisms are boosted with machine-learned advice. The problem is surely well-motivated, considering that small and large advertisers simultaneously exist in the market.
- The theoretical results given by the paper are solid, which include the tight performance of VCG, the optimality of VCG within a broad class of truthful auctions, and the extension to GSP and GFP.
- This paper conducts experiments to corroborate their theoretical results.
- This paper is well-written.

Cons:
- My major concern with the paper is the individual welfare metric. I am not fully aware of the underlying logic. Here, the metric fixes an advertiser's bidding strategy and has other bidders satisfy that advertiser's ROAS constraint. I find this a bit hard to interpret. Intuitively, the ROAS constraint should be satisfied by the auction mechanism rather than other advertisers, especially considering that the mechanism indeed knows partial information on advertisers' values (the ML advice). However, it is certain (as pointed out by the authors) that even with the above information, a single advertiser's welfare under any mechanism can be zero with some special problem instances. I hope the authors could elaborate on this dilemma and their metric choice. It seems to me that defining the metric under some kind of market equilibrium could be a better trial. In fact, in the details of your experiment (Appendix F.1), you use gradient descent to have the bid multipliers converge to an "equilibrium", which seems to match the metric I am talking about above.
- I am also afraid that the ML advice adopted by this paper is a bit too strong, especially in the sense that the advice should have a good approximation of every value.

**Questions:**

1. Could you elaborate more on your metric choice, which I mentioned above in the cons?
2. As a continuation of the model choice, could we have better guarantees when we add some certain assumptions, e.g., each auction is well-competed (as opposed to the motivating example in Section 3.1), or distinguish between large and small advertisers and consider their individual welfare guarantee respectively? Such kind of results could be meaningful.
3. In Appendix F.1, when you update the multipliers, you do not use the reserves for the first $T$ rounds, but only include them in the last $T$ rounds. Could you explain why?

**Reviewer Confidence:**

3: The reviewer is confident but not certain that the evaluation is correct

**Scope:**

4: The work is relevant to the Web and to the track, and is of broad interest to the community

---

### Official Review · Reviewer_L7ri · 2023-11-23

**Novelty:** 5
**Technical Quality:** 4

**Review:**

To maximize total advertiser welfare to improve overall channel, previous research has explored using machine learning predictions on advertiser values (referred to as machine-learned advice) to enhance total welfare. The draft recognizes that such improvements may negatively impact individual bidders' welfare and doesn't address how specific advertiser bidding strategies influence welfare.
Pros:
1. Overall, this draft is well-written, and well structured.
2.The draft proposed a worst-case welfare lower-bound guarantee for an individual auto-bidder, and gives  a deeper and principled understanding to this problem.

Cons:
1.Code is not shared which makes replicates of the experiments not easy.
2. The empirical evaluation of the proposed method is not enough. Firstly, the draft did not introduce other baselines, which makes it hard to evaluate the proposed method.
3. The author claimed that the main contribution of this work is that it can guarantee a worst-case welfare lower-bound guarantee for an individual auto-bidder. It would be interesting how severe will existing algorithms ignore the welfare for an individual auto-bidder. It would be better to have experiments to show it since this is the main motivation of this draft.

**Questions:**

See the above review.

**Reviewer Confidence:**

2: The reviewer is willing to defend the evaluation, but it is likely that the reviewer did not understand parts of the paper

**Scope:**

3: The work is somewhat relevant to the Web and to the track, and is of narrow interest to a sub-community

---

### Official Review · Reviewer_FH9E · 2023-11-24

**Novelty:** 6
**Technical Quality:** 6

**Review:**

The paper analyses welfare loss for individual bidders in the autobidding world for auctions with and without machine-learned advice. They study how ML predictions on advertiser values can be used to improve individual welfares.
They present a new individual welfare guarantee metric for individual advertisers that depends on the strategy. They further show that using  ML advice improves individual welfare guarantees under their proposed individual welfare metric. They show some guarantees on VCG and extend their work to GSP setting. Further they provide numerical results using semisynthetic data derived from the auction logs of a search ad platform. They show as the predictions improve in quality, the welfare approaches that of efficient outcomes.

The paper is generally well written and well structured. The results derived are interesting and insightful. However, the empirical section could be made stronger. It would be interesting to see some ML models in action, predicting values based on features, and how that improves the auction results.

**Questions:**

1. Have the authors experimented with ML models to predict the values, instead of sampling from a distribution with a specified accuracy value (\beta)?
2. Do you envision a system with a feedback loop, where the predictions of such an ML model can improve with feedback from the auction results, and thereby showing a positive reinforcement on the auction outcomes as well?

**Reviewer Confidence:**

2: The reviewer is willing to defend the evaluation, but it is likely that the reviewer did not understand parts of the paper

**Scope:**

4: The work is relevant to the Web and to the track, and is of broad interest to the community

---

### Official Review · Reviewer_f34X · 2023-12-01

**Novelty:** 5
**Technical Quality:** 5

**Review:**

The paper focuses on the problem of Auto-bidding with machine-learned advice.
This work theoretically demonstrates how incorporating ML advice as reserves in classic auctions (e.g., VCG, GSP, GFP) can improve individual bidder welfare.
The authors present numerical studies using semi-synthetic data from ad auction logs of a search ad platform to show improvements in individual welfare when setting personalized reserve prices with ML advice.


(+) The paper targets a practical problem.

(+) The related work section is detailed and well-explained.

(-) There is no experimental evaluation using real datasets.

(-) The definition of ML advice / ML algorithms is somewhat unclear.


The paper addresses an important issue.
It provides a good brief survey of the related work.
It is well-motivated, and its approach seems interesting and reasonable.

The definition of ML advice / ML algorithms is somewhat unclear.
What kind of ML algorithms should be considered?
I think providing some short explanations or running examples in the introduction would be helpful.

Also, there is no experimental evaluation using real datasets.
I would like to see some empirical evidence to demonstrate the effectiveness of the proposed solution.

**Questions:**

N/A

**Reviewer Confidence:**

2: The reviewer is willing to defend the evaluation, but it is likely that the reviewer did not understand parts of the paper

**Scope:**

3: The work is somewhat relevant to the Web and to the track, and is of narrow interest to a sub-community

---

### Decision · Program_Chairs · 2024-01-22

**Decision:**

Accept (Oral)

**Comment:**

Summary: The paper analyses welfare loss for individual bidders in the autobidding world for auctions with and without machine-learned advice.

 Strengths:
 + well-written and well-structured
 + targets an interesting and important problem
 + good discussion of related literature

 Weaknesses:
 - lack of empirical validation on real datasets
 - some discussion needed on the nature and strength of ML advice

 Recommendation: Accept. Solid theoretical work, with practical applicability to be accentuated.